# Unveiling Perceptual Artifacts: A Fine-Grained Benchmark for Interpretable AI-Generated Image Detection

**Yao Xiao**[1*]**, Weiyan Chen**[1*]**, Jiahao Chen**[3*]**, Zijie Cao**[1*]**, Weijian Deng**[4]**, Binbin Yang**[1]**,
**Ziyi Dong**[1]**, Xiangyang Ji**[5]**, Wei Ke**[3]**, Pengxu Wei**[1,2†]**, Liang Lin**[1,2]

[1]Sun Yat-sen University, [2]Peng Cheng Laboratory, [3]Xi'an Jiaotong University,
[4]Australian National University, [5]Tsinghua University
xiaoy99@mail2.sysu.edu.cn, weipx3@mail.sysu.edu.cn

## Abstract

Current AI-Generated Image (AIGI) detection approaches predominantly rely on binary classification to distinguish real from synthetic images, often lacking interpretable or convincing evidence to substantiate their decisions. This limitation stems from existing AIGI detection benchmarks, which, despite featuring a broad collection of synthetic images, remain restricted in their coverage of artifact diversity and lack detailed, localized annotations. To bridge this gap, we introduce a fine-grained benchmark towards eXplainable AI-Generated image Detection, named X-AIGD, which provides pixel-level, categorized annotations of perceptual artifacts, spanning low-level distortions, high-level semantics, and cognitive-level counterfactuals. These comprehensive annotations facilitate fine-grained interpretability evaluation and deeper insight into model decision-making processes. Our extensive investigation using X-AIGD provides several key insights: (1) Existing AIGI detectors demonstrate negligible reliance on perceptual artifacts, even at the most basic distortion level. (2) While AIGI detectors can be trained to identify specific artifacts, they still substantially base their judgment on uninterpretable features. (3) Explicitly aligning model attention with artifact regions can increase the interpretability and generalization of detectors. The data and code are available at: https://github.com/Coxy7/X-AIGD.

## 1 Introduction

As AI-Generated Images (AIGIs) become increasingly indistinguishable from real photographs, there is a growing demand for detection methods that are not only accurate but also interpretable (Li et al., 2024a; Wen et al., 2025; Gao et al., 2025). Traditional detection approaches typically frame AIGI detection as a binary classification task, leveraging handcrafted or learned features (Ojha et al., 2023; Tan et al., 2024; Koutlis & Papadopoulos, 2024; Chen et al., 2024a;b). While effective on specific datasets, these methods often struggle to generalize across different image generators and are vulnerable to perturbations in image structure and content. More importantly, their reliance on low-level signals offers limited interpretability, providing only binary decisions without identifying specific visual artifacts for decision making, which is unconvincing for practical applications.

To enhance interpretability, a series of research (Huang et al., 2024; Xu et al., 2025; Liu et al., 2024c; Wen et al., 2025; Gao et al., 2025) attempts to utilize the visual reasoning capabilities of Multimodal Large Language Models (MLLMs) to generate textual explanations for AIGI detection. However, due to the lack of human annotations of detection clues, these works primarily rely on data annotated by stronger MLLMs such as GPT-4o (OpenAI, 2024) for model training, which may be unreliable and lack spatial grounding. Recent works (Ye et al., 2025; Kang et al., 2025) have made progress by providing human-annotated datasets with localized artifact annotations, following previous studies on perceptual artifact localization (Zhang et al., 2023; Cao et al., 2024). However, these datasets lack systematic and fine-grained categorization of artifacts, limiting the depth of interpretability analysis.

---

*Equal contribution. †: Corresponding author.

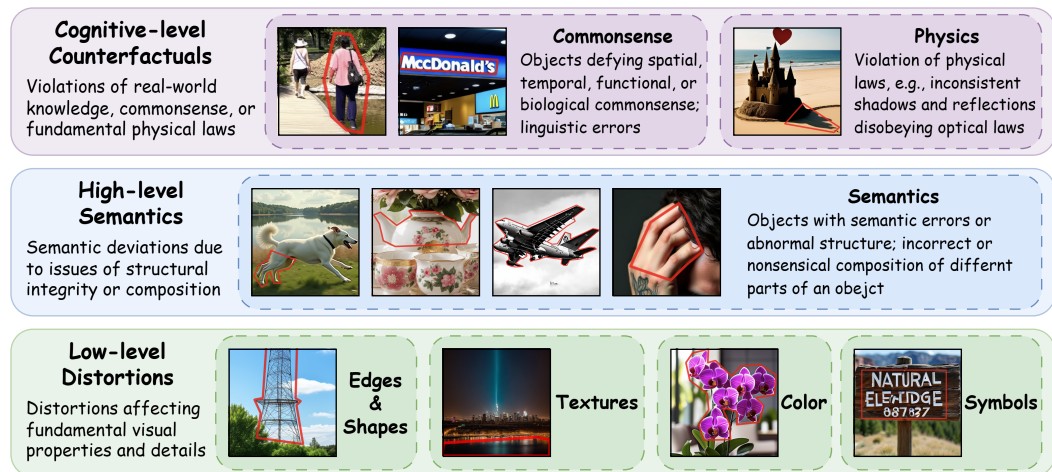

Figure 1: **Taxonomy of Perceptual Artifacts.** Our X-AIGD categorizes perceptual artifacts into three hierarchical levels: 1) low-level distortions, 2) high-level semantics, and 3) cognitive-level counterfactuals. Low-level distortions capture fundamental visual irregularities, such as edge misalignments, unnatural textures, and color inconsistencies. High-level semantics address structural and compositional errors that disrupt object integrity and logical arrangement. Cognitive-level counterfactuals encompass commonsense and physical violations that defy real-world knowledge, such as implausible object relationships or violations of physical laws. Red polygons illustrate the precise localization of these artifacts, supporting fine-grained evaluation of AIGI detection models.

Consequently, the development of robust and interpretable AIGI detection methods remains hindered by the lack of reliable and comprehensive datasets that capture the full perceptual artifacts in AIGIs.

To address these limitations, we introduce a novel benchmark, named X-AIGD (e**X**plainable **AI-G**enerated image **D**etection), to elevate the interpretability and robustness of AIGI detection. X-AIGD features pixel-level annotations that span a diverse range of perceptual artifacts, systematically categorized into three levels: low-level distortions (*e.g.*, unnatural textures, warped edges), high-level semantics (*e.g.*, abnormal object structures), and cognitive-level counterfactuals (*e.g.*, violations of commonsense or physical laws), as illustrated in Fig. 1. This structured taxonomy captures not only easily observable distortions but also deep semantic inconsistencies and logical violations, offering a more comprehensive evaluation of model capabilities. For reliable assessment of interpretability and detection performance, we collect high-quality, diverse fake images generated from the captions of real images by advanced generative models (*e.g.*, FLUX (Labs, 2024) and Stable Diffusion 3.5 (Esser et al., 2024)), ensuring semantic alignment and distribution consistency between fake and real image pairs. As summarized in Tab. 1, X-AIGD provides paired real and fake images, along with pixel-level annotations and systematic artifact categorization, making it well-suited to support research on interpretable AIGI detection.

Leveraging the fine-grained annotations and paired real and fake images in X-AIGD, we conduct a multi-faceted investigation into whether and how AIGI detection models can utilize perceptual artifacts as reliable, interpretable cues. We begin by analyzing existing end-to-end AIGI detectors to assess their reliance on perceptual artifacts. Despite their ability to learn diverse features, it is observed that they commonly bypass perceptual artifacts. This underutilization of human-interpretable cues suggests an opportunity for improving detection mechanisms by more effectively incorporating artifact-based reasoning.

The limitations observed in binary classification motivate us to investigate whether explicitly detecting perceptual artifacts could enhance both detection performance and interpretability. Utilizing the annotated images in X-AIGD, we explore several approaches that adopt artifact detection as an auxiliary task to guide authenticity judgment, including transfer learning and joint training. While these models show promise in identifying certain low-level artifacts, their authenticity predictions still predominantly depend on uninterpretable features, and the performance gains from artifact detection are at most marginal. This observation underscores the challenge of aligning detection decisions with transparent, artifact-based reasoning.

To further explore the potential of artifact-based reasoning, we investigate whether explicitly aligning model attention with artifact regions can enhance the interpretability and generalization of AIGI detectors. By incorporating attention alignment loss during training, we observe improved focus on artifact regions and significantly enhanced performance in cross-dataset evaluations. Further trade-off analysis on the penalty of attention outside artifact regions reveals that balancing the learning of perceptual artifacts with other useful features is crucial for optimal performance in practice. Overall, our findings highlight the potential of artifact-aware detection and the complexity of developing AIGI detectors that are both accurate and interpretable, emphasizing the need for continued research into AIGI detection methods that effectively leverage perceptual artifacts.

Our primary contributions are summarized as follows:

- We propose X-AIGD, a fine-grained benchmark designed to facilitate in-depth exploration of interpretable AIGI detection, featuring localized, multi-level artifact annotations.

- Our analysis reveals that existing AIGI detectors underutilize perceptual artifacts, hindering interpretability and detection robustness. Further, we find that detectors trained with artifact detection as an auxiliary task still rely heavily on uninterpretable features, highlighting the challenge of aligning detection with perceptual cues.

- We demonstrate that aligning model attention with artifact regions can enhance both interpretability and generalization, underlining the potential of artifact-aware detection methods.

## 2 RELATED WORKS

**AI-Generated Image Detection.** Existing AIGI detection datasets proposed for the binary classification scenario (Zhu et al., 2023; Bammey, 2023; Baraldi et al., 2024; Chen et al., 2024a) typically comprise paired real and fake images and cover a variety of generative models, focusing on the evaluation of the generalization ability of AIGI detectors. Based on these datasets, most previous studies either adopt end-to-end models (Wang et al., 2020b; Ojha et al., 2023; Koutlis & Papadopoulos, 2024; Chen et al., 2024a; Liu et al., 2024b), or propose fingerprint-based methods (Wang et al., 2023; Tan et al., 2024; Zhong et al., 2024; Chen et al., 2024b) that rely on certain low-level synthetic traces such as the up-sampling artifacts (Tan et al., 2024). Despite their reported high accuracy, the interpretability of these models is not well-studied due to the lack of datasets with comprehensive and grounded annotations of the visual clues for AIGI detection.

**Benchmarks for interpretable AIGI detection.** To support the research on interpretable AIGI detection, existing works (Li et al., 2024a; Gao et al., 2025; Wen et al., 2025) have primarily focused on MLLMs and introduced benchmark datasets with textual annotations providing interpretations of artifacts. However, their interpretability evaluation often involves comparing the models' textual outputs against ground truth text at the image level, lacking grounding to specific anomaly regions. This undermines the efficacy of the evaluation, as it prevents verifying whether the model's interpretation is truly based on relevant visual evidence. LOKI (Ye et al., 2025) and SynthScars (Kang et al., 2025) provide localized annotations of image anomalies based on human-annotated bounding boxes or pixel masks. Nevertheless, they treat artifact detection as a separate task from AIGI detection with minimal examination of the their synergy. Moreover, the lack of paired real images limits their applicability for developing and evaluating interpretable AIGI detectors.

**Perceptual Artifact Localization for Generated Images.** Previous studies related to perceptual artifact localization (Zhang et al., 2023; Cao et al., 2024; Liang et al., 2024; Fang et al., 2024; Wang et al., 2025) aim to construct artifact localization models that provide feedback for tuning generative models or regional guidance for repairing generated images. Therefore, their collected datasets mainly focus on specific artifacts that deteriorate the visual quality. In contrast, our X-AIGD aims at the comprehensive evaluation and fine-grained interpretability analysis of AIGI detectors, thus includes a wider range of artifacts that can serve as detection clues, and provides paired real images.

| Dataset | Real Images | Artifact Annotation | |
| --- | --- | --- | --- |
| | | Localization | #Category |
| PAL4VST (Zhang et al., 2023) | ✗ | Pixel mask | N/A |
| SynArtifact (Cao et al., 2024) | ✗ | Bounding box | 13 |
| RichHF-18K (Liang et al., 2024) | ✗ | Point | N/A |
| LOKI (image) (Ye et al., 2025) | Unpaired | Bounding box | N/A |
| FakeBench (Li et al., 2024a) | Unpaired | N/A | 14 |
| MMFR-Dataset (Gao et al., 2025) | Paired | N/A | 5 |
| SynthScars (Kang et al., 2025) | ✗ | Pixel mask | 3 |
| X-AIGD (ours) | Paired | Pixel mask | 7 |

Table 1: **Comparison with existing datasets.** X-AIGD provides detailed annotations (pixel-level masks and categorized artifact labels), real images paired with the fake ones, supporting fine-grained evaluation. PAL4VST, SynArtifact, and RichHF-18K are for perceptual artifact localization.

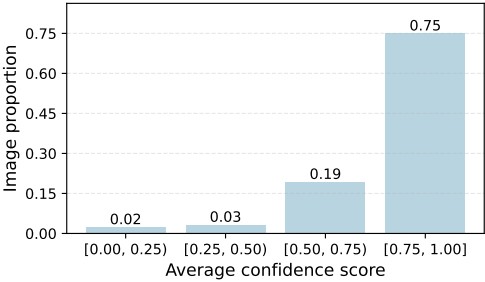

Figure 2: **Human assessment of annotation quality.** The distribution of average confidence scores assigned by human annotators indicates the overall high quality of artifact annotations.

# 3 X-AIGD Benchmark

## 3.1 Image Collection

The reliable and comprehensive evaluation of the interpretability of AIGI detectors requires a high-quality and diverse dataset of real and fake images, where a reasonable number of perceptual artifact instances can be identified and localized for most fake images. To this end, we collect fake images depicting realistic scenes based on the semantics of real images from various sources. Specifically, we source real images from four existing datasets, and obtain captions of these images with different levels of detail, which serve as a diverse set of text prompts for fake image generation. To ensure the quality of the generated images, we use 13 up-to-date text-conditioned image generation models (*e.g.*, PixArt-$\alpha$ (Chen et al., 2024c), FLUX.1-dev (Labs, 2024), and SD 3.5 (Esser et al., 2024)). We also include community fine-tuned models on Civitai (Civ) that aims at improving generation realism. Prompt engineering (Liu & Chilton, 2022) techniques are applied to further enhance image quality and suppress non-realistic styles following (Baraldi et al., 2024). In total, we collect 4,000 real images and generate 4,000 corresponding fake images from each of the 13 generators, resulting in 52,000 fake images. Additional details are provided in Appendix A.

Paired real and fake images with semantic alignment are important for assessing models' ability to base their judgment on localized perceptual artifacts instead of merely the overall semantics of the images. However, as compared in Tab. 1, the real images provided by existing datasets for interpretable AIGI detection (*i.e.*, LOKI (Ye et al., 2025), FakeBench (Li et al., 2024a)) are not paired with their fake images. Furthermore, datasets proposed for perceptual artifact localization (Zhang et al., 2023; Cao et al., 2024; Liang et al., 2024) do not provide real images and therefore are less suitable for the study on interpretable AIGI detection. A more comprehensive comparison of related datasets is presented in Tab. 8.

## 3.2 Annotation of Perceptual Artifacts

For reliable interpretability assessment, ground truth annotations should be complete and accurately capture perceptual artifacts of all relevant types perceivable by humans. To this end, we define a comprehensive artifact taxonomy comprising 3 levels and 7 specific categories, as shown in Fig. 1. We recruit 12 human annotators to label artifacts in fake images by localizing them with pixel-level polygon masks and assigning the corresponding category according to our taxonomy. To improve annotation completeness, each image undergoes three successive rounds of annotation by different annotators. Additionally, we exclude images flagged as low-quality or unrealistic by annotators.

Given the labor-intensive nature of annotation, we select a subset of the fake images collected in Sec. 3.1 for detailed annotation. Specifically, the test set comprises 200 fake images for each of the 13 generators, along with their corresponding real images. The training set is constructed similarly using images from a subset of 5 generators. Following annotation and filtering, we obtain a total of 3,035 valid annotated fake samples. To evaluate the annotation quality, we randomly sample a subset of annotated fake samples and ask three independent annotators to assign a confidence score

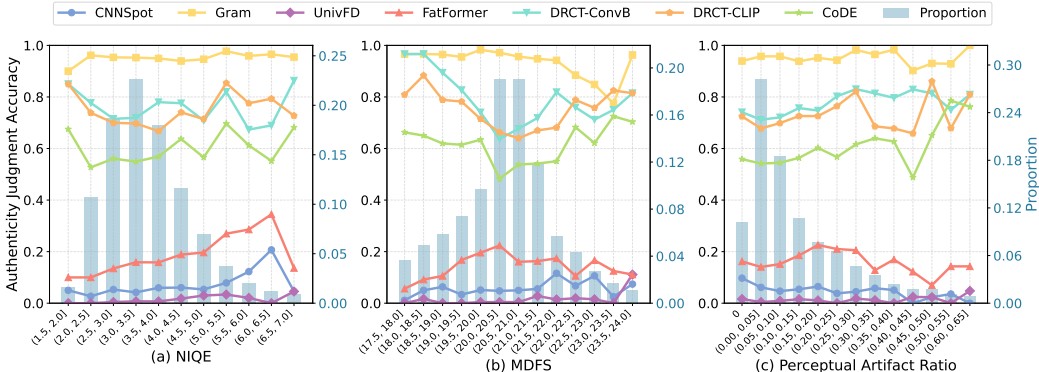

Figure 3: **Accuracy of existing AIGI detectors on AIGIs with different fidelity.** The image fidelity is measured by (a) NIQE (Mittal et al., 2012); (b) MDFS (Ni et al., 2024); (c) Perceptual Artifact Ratio (Zhang et al., 2023) computed based on our annotations. For these three metrics, higher values indicate lower fidelity.

from $0, 0.5, 1$ to each annotated artifact instance. The distribution of the average confidence scores is presented in Fig. 2, indicating generally high annotation quality, despite a small proportion of controversial instances due to the subjective nature of the human perception of certain artifacts.

## 3.3 TASK SPECIFICATION

The X-AIGD dataset is designed for interpretable AIGI detection, a task that requires models to provide interpretations for their judgment of image authenticity. Specifically, we focus on grounded interpretations, which provide localized evidence (*i.e.*, artifact regions) and categorical explanations (*i.e.*, artifact types) understandable to humans. Formally, interpretable AIGI detection can be divided into two subtasks: **(1) Authenticity Judgment (AJ)**: Predict the binary label $y \in \{0, 1\}$, where $y = 1$ suggests a fake image generated by AI, and $y = 0$ indicates the image is a real-world photograph. For this binary classification task, we adopt balanced accuracy (**Acc**), defined as the average of real-image accuracy and fake-image accuracy, along with precision (**P**), recall (**R**), and $F_1$-score (**$F_1$**), to assess the model performance. **(2) Perceptual Artifact Detection (PAD)**: If the image is predicted to be fake, detect a set of perceptual artifact instances. For each detected instance $i$, predict its region $r_i$ and category $c_i \in C$. The set of categories $C$ includes 7 types of perceptual artifacts spanning low-level distortions, high-level semantics, and cognitive-level counterfactuals, as illustrated in Fig. 1. To assess the models' ability in detecting artifacts of a specific category, we compare the model predictions with ground-truth artifact regions of this category, and calculate the Intersection over Union (**IoU**), as well as pixel-level precision, recall, and $F_1$-score (**PixP**, **PixR**, and **$\text{PixF}_1$**). In addition, we propose alternative instance-level weak-localization metrics, which are compatible with a wider range of models and are noise tolerant, as discussed in Sec. A.5. To accommodate the evaluations of models unaware of the artifact categories of X-AIGD, we propose a **Category-Agnostic PAD** setting, which focuses on the existence of perceptual artifacts without considering the categories. Specifically, for each image, we obtain a ground-truth binary mask from the union of annotations of all categories, and compute the same metrics as PAD based on the predicted binary mask of artifact regions.

## 4 ASSESSING THE INTERPRETATION OF EXISTING AIGI DETECTORS

Despite the performance of existing AIGI detectors on previous benchmark datasets (Wang et al., 2020b; Chen et al., 2024a; Bammey, 2023), it is unclear whether they can detect any kind of perceptual artifacts, given that they are not explicitly designed to align their detection clues with human perception. In this section, we conduct quantitative and qualitative analyses based on X-AIGD to assess their ability to detect perceptual artifacts.

**AIGI detectors.** Our interpretability investigation focuses on end-to-end expert AIGI detection models, which potentially learn a wide range of features for AIGI detection, including perceptual artifacts. We evaluate the following pre-trained models: CNNSpot (Wang et al., 2020b), Gram-

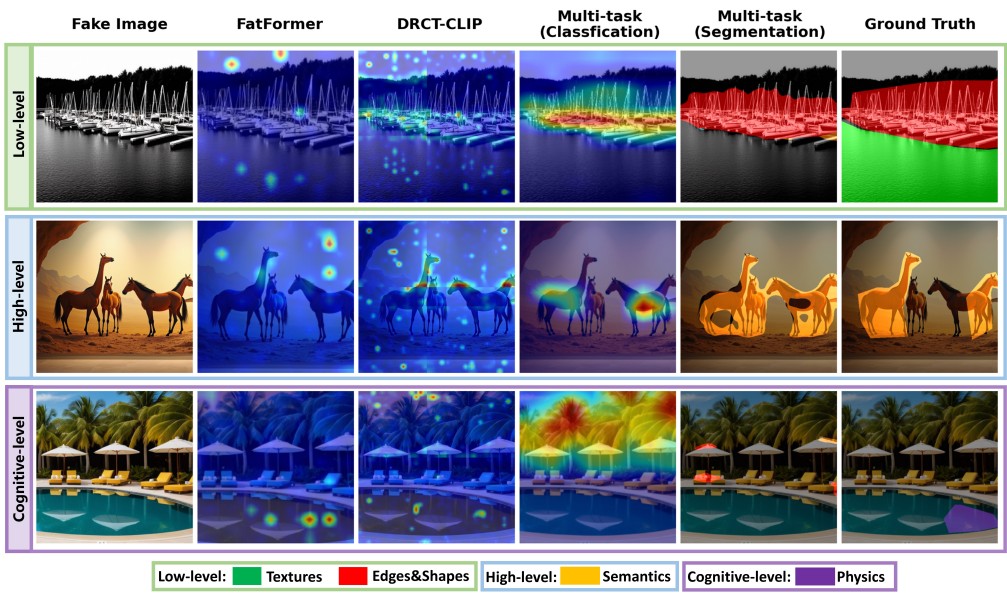

Figure 4: **Qualitative comparison of the interpretations of different models.** Columns 2-4 show the interpretation heatmap for existing AIGI detectors (FatFormer (Liu et al., 2024b) and DRCT-CLIP (Chen et al., 2024a)) and the classification head of the multi-task model. Column 5 is the artifact segmentation results of the multi-task model. The last column shows the ground truth annotations.

Net (Liu et al., 2020), UnivFD (Ojha et al., 2023), FatFormer (Liu et al., 2024b), DRCT-CLIP (Chen et al., 2024a), DRCT-ConvB (Chen et al., 2024a), and CoDE (Baraldi et al., 2024). Further details are presented in Appendix B.1. In addition, we provide a dedicated evaluation of MLLMs and related discussions in Appendix C.

**Relationship between authenticity judgment accuracy and image fidelity.** Intuitively, AIGIs with lower fidelity or more perceptual artifacts should be easier to detect if models effectively leverage these artifacts as cues. To explore this, we evaluate existing AIGI detectors on samples with varying fidelity levels from X-AIGD test set, measured by NIQE (Mittal et al., 2012), MDFS (Ni et al., 2024), and Perceptual Artifact Ratio (PAR) (Zhang et al., 2023). While NIQE and MDFS are no-reference quality metrics, PAR quantifies the proportion of artifact region in images based on our annotations. Interestingly, as shown in Fig. 3, the detector accuracy does not have significant correlation with image fidelity. Moreover, models do not consistently perform better on AIGIs with visible artifacts (PAR > 0) compared to those without (PAR = 0), indicating their limited utilization of perceptual cues for detection.

**Quantitative evaluation on model interpretations.** To evaluate the potential capability of models in detecting various types of perceptual artifacts, a quantitative analysis is conducted based on the interpretation heatmaps for the fake class as model interpretations. We apply Grad-CAM (Selvaraju et al., 2017) to CNN-based models and Relevance Map (Chefer et al., 2021) to Vision Transformers (Dosovitskiy et al., 2021). For patch-based models (DRCT and CoDE), heatmaps are generated through a sliding-patch approach inspired by Chai et al. (2020). The heatmaps are binarized with a threshold of 0.5 and compared against the ground-truth artifact masks. The results in Category-Agnostic PAD (Tab. 2) indicate weak alignment between model interpretations and human perception.

**Qualitative analysis.** We visualize the interpretation heatmaps of two representative models, FatFormer (Liu et al., 2024b) and DRCT-CLIP (Chen et al., 2024a), alongside the ground-truth masks of perceptual artifacts in Fig. 4. It is shown that FatFormer primarily concentrates on background regions with smooth textures, while DRCT-CLIP exhibits a more scattered activation pattern with heightened focus on edges. These qualitative results indicate that neither model have consistent attention to perceptual artifacts, reflecting limitations in artifact-aware detection. Additional examples are presented in Fig. 12.

Table 2: **Comparison of existing AIGI detectors and the models trained on our dataset on** *Authenticity Judgment* **and** *Category-Agnostic PAD.* For existing models, the binarized interpretation heatmaps of the classifiers are used for the evaluation on Category-Agnostic PAD. Training on our artifact annotations, models achieve non-trivial performance on Category-Agnostic PAD, but transfer learning from PAD to AJ and multi-task learning yield at most marginal improvements in AJ.

| Model | Authenticity Judgment | | | | Category-Agnostic PAD | | | |
|---|---|---|---|---|---|---|---|---|
| | Acc | P | R | $F_1$ | IoU | PixP | PixR | PixF$_1$ |
| CNNSpot (Wang et al., 2020b) | 48.6 | 39.7 | 5.6 | 9.8 | 0.9 | 7.7 | 1.0 | 1.8 |
| Gram-Net (Liu et al., 2020) | 62.7 | 57.7 | 95.0 | 71.8 | 5.8 | 16.9 | 8.1 | 11.0 |
| UnivFD (Ojha et al., 2023) | 50.2 | 64.1 | 1.0 | 2.0 | 0.4 | 10.1 | 0.4 | 0.8 |
| FatFormer (Liu et al., 2024b) | 52.1 | 57.3 | 16.3 | 25.4 | 0.5 | 10.2 | 0.5 | 0.9 |
| DRCT-ConvB (Chen et al., 2024a) | 82.5 | 88.4 | 74.8 | 81.0 | 9.0 | 14.8 | 18.8 | 16.6 |
| DRCT-CLIP (Chen et al., 2024a) | 53.0 | 52.2 | 71.4 | 60.3 | 0.7 | 13.7 | 0.8 | 1.5 |
| CoDE (Baraldi et al., 2024) | 76.5 | 93.1 | 57.2 | 70.9 | 2.9 | 11.5 | 3.8 | 5.7 |
| AJ-only | 89.3 | **96.3** | 85.8 | 90.2 | / | / | / | / |
| PAD-only | / | / | / | / | 27.2 | **40.4** | 45.4 | 42.7 |
| Transfer learning (linear probing) | 82.1 | 93.5 | 73.9 | 82.5 | / | / | / | / |
| Transfer learning (full fine-tuning) | **89.9** | 93.5 | 91.6 | **92.5** | / | / | / | / |
| Multi-task learning | 89.1 | 92.6 | **92.0** | 92.3 | **27.3** | 36.2 | **52.5** | **42.8** |

Table 3: **Fine-grained evaluation on** *Perceptual Artifact Detection*. The multi-task and single-task (PAD-only) models perform better on low-level artifacts but struggle with high-level ones.

| Model | Low-level | | | | | | | | High-level | | Cognitive-level | | | |
|---|---|---|---|---|---|---|---|---|---|---|---|---|---|---|
| | Textures | | Edges&Shapes | | Symbols | | Color | | Semantics | | Commonsense | | Physics | |
| | PixP | PixR | PixP | PixR | PixP | PixR | PixP | PixR | PixP | PixR | PixP | PixR | PixP | PixR |
| PAD-only | **17.7** | 9.1 | **32.8** | 50.1 | **43.8** | **54.2** | **10.9** | 2.5 | **19.9** | 23.5 | **11.3** | 2.2 | 2.9 | 0.3 |
| Multi-task | 14.9 | **15.6** | 29.8 | **55.6** | 43.3 | 52.3 | 8.4 | **5.4** | 17.9 | **25.5** | 7.4 | 2.2 | **3.4** | **0.8** |

# 5 EXPLORING PERCEPTUAL ARTIFACT DETECTION FOR INTERPRETABLE AIGI DETECTION

Given the limited interpretability of existing AIGI detectors, it is natural to ask whether more interpretable models can be developed by explicitly detecting perceptual artifacts to justify their decisions. Although Zhang et al. (2023) and Kang et al. (2025) have demonstrated the feasibility of training segmentation models to detect certain types of perceptual artifacts, it remains unexplored whether AIGI detectors can effectively utilize the perceptual artifacts as reliable and interpretable cues for authenticity judgment. To explore this, we employ PAD as an auxiliary task to guide authenticity judgment via transfer learning or multi-task learning, and analyze the model behavior on both tasks.

**PAD as an auxiliary task for AJ.** To investigate whether PAD can enhance the interpretability and performance of AIGI detectors, we explore two approaches: *1) Transfer learning*, where a segmentation model is first pre-trained on PAD and then used to initialize the backbone of a binary classification model for linear probing or full fine-tuning on AJ; *2) Multi-task learning*, where a single model is jointly trained on both tasks. For unified comparison, we use the same Swin Transformer (Liu et al., 2021) as the backbone for all models. Details are provided in Appendix B.2.

**Fine-grained evaluation on PAD.** To validate that both the PAD-only model (*i.e.*, the pre-trained model used for transfer learning) and the multi-task model effectively learn from the human annotations, we compare their segmentation results with the interpretations of existing AIGI detectors. The Category-Agnostic PAD performance in Tab. 2 shows that both models achieve significantly better results than existing models, demonstrating their non-trivial ability to detect perceptual artifacts. Furthermore, to understand their strengths and weaknesses in detecting different types of perceptual artifacts, we report category-specific results in Tab. 3, and qualitative visualizations in Fig. 4. First, for low-level artifacts, the models perform relatively well in detecting distorted edges, shapes, and symbols, as indicated by the quantitative results. Second, while some high-level artifacts are detected, the models tend to associate certain objects or parts with structural errors (e.g., the horses in the second row of Fig. 4), and struggles to differentiate semantically correct instances from incorrect ones. Third, the model exhibits clear limitations in detecting cognitive-level artifacts, such as incorrect reflections (e.g., the third row of Fig. 4), which highlights the inherent challenges of traditional vision models in reasoning about commonsense and physical consistency. Overall, despite promising results

for certain artifact types, the models fall short in generalizing across the full spectrum of perceptual artifacts, underscoring the complexity of the PAD task and the gap in interpretability.

**Case study: text rendering artifacts.** To further demonstrate the PAD capability of the multi-task model, we conduct a case study on text rendering artifacts, which are common in AIGIs. According to our artifact taxonomy, text rendering issues span both low-level distortions and cognitive-level counterfactuals. Distorted letters are detectable without understanding word spelling or meaning, whereas legible texts require linguistic and commonsense knowledge to identify spelling mistakes or nonsensical expressions. The qualitative examples in Fig. 5 illustrate that the multi-task model effectively identifies distorted letters in case (a) but produces false alarms for legible texts in case (d). As anticipated, it struggles with spelling errors in case (b). Interestingly, the model detects some nonsensical words as commonsense violations in case (c), yet the predicted masks are intermingled with low-level artifacts, indicating its limited capacity to differentiate between types of text-related issues.

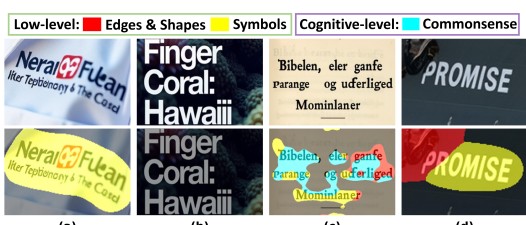

Figure 5: **PAD results of multi-task model on different types of text rendering artifacts:** (a) distorted letters; (b) spelling mistakes; (c) nonsensical words; (d) no evident issues.

**Analyzing the effect of auxiliary PAD learning on AJ.** To investigate whether learning the PAD task can enhance the interpretability and performance of AIGI detectors, we first compare the transfer learning and multi-task models with the single-task baseline (*i.e.*, AJ-only). As suggested by the comparisons in Tab. 2, linear probing after PAD pre-training leads to significantly lower recall, indicating that the features learned from PAD alone are insufficient for effective AJ. Full fine-tuning and multi-task learning yield slightly better $F_1$-scores than the AJ-only model, suggesting a limited positive effect of PAD on AJ. Furthermore, to assess how the multi-task model may depend on the perceptual artifacts for authenticity judgment, we compare the classification Grad-CAM heatmaps and the segmentation outputs in Fig. 4. Despite certain overlaps on low-level artifact regions, the heatmaps show high activations on some background areas of the segmentation. In other words, the model still tends to rely on uninterpretable features for authenticity judgment, and the perceptual artifacts detected by the segmentation head are not necessarily the main clues for the model's judgment. Additional quantitative evaluation of interpretations is presented in Appendix D.3.

# 6 IMPROVING AIGI DETECTORS WITH EXPLICIT ATTENTION ALIGNMENT

## 6.1 ALIGNING MODEL ATTENTION WITH PERCEPTUAL ARTIFACT REGIONS

As discussed in Sec. 5, using PAD as an auxiliary task does not effectively encourage AIGI detectors to focus on perceptual artifacts for authenticity judgment, and the model attention may still be distracted by uninterpretable features. This motivates us to explore a more direct way that regularizes model attention to align with perceptual artifact regions.

To this end, we propose an attention alignment method that can be applied to ViT-based AIGI detectors. Specifically, we employ the Gradient Attention Rollout (Gildenblat, 2020), an extension of Attention Rollout (Abnar & Zuidema, 2020) that computes the aggregated attention heatmap $A_{cls} \in [0,1]^{h \times w}$ for the classification logits with respect to all $h \times w$ image patches. This heatmap reflects the model's attention distribution for predicting the image as fake, and is compared with the annotated artifact regions. To match the patch-level attention heatmap, we construct a corresponding patch-level artifact heatmap $A_{art} \in [0,1]^{h \times w}$ by setting each value $A_{art}^{(i,j)}$ as the proportion of pixels belonging to any annotated artifact within the corresponding patch. For real images, we set $A_{art}$ as a zero matrix. Since the computation of $A_{cls}$ is differentiable, we can optimize the mean squared error between the two heatmaps as an auxiliary loss during training. However, restricting the model attention to only artifact regions may hinder the model from learning other useful features that appear in benign regions (*i.e.*, where $A_{art}^{(i,j)} = 0$). Therefore, we introduce a *benign region weight* $\lambda \in [0,1]$

Table 4: **Ablation study on attention alignment.** Artifact-based alignment generally achieves superior AJ performance to baseline without alignment ($\beta = 0$) and saliency-based alignment across datasets.

| Alignment | X-AIGD | | Synthbuster | | Chameleon | | CommFor | |
|---|---|---|---|---|---|---|---|---|
| Mask | Acc | $F_1$ | Acc | $F_1$ | Acc | $F_1$ | Acc | $F_1$ |
| None ($\beta = 0$) | 85.7 | 84.3 | 69.1 | 55.9 | **71.4** | 62.7 | 69.7 | 58.2 |
| Saliency | 86.6 | 86.4 | 71.2 | 60.7 | 68.4 | 58.3 | 69.2 | 59.7 |
| Artifact | **87.1** | **87.4** | **72.1** | **63.2** | 71.2 | **63.5** | **69.8** | **61.4** |

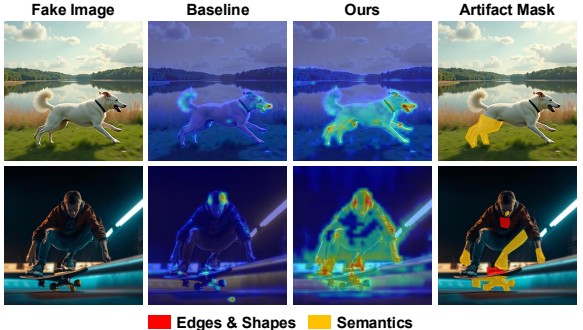

**Fake Image**  **Baseline**  **Ours**  **Artifact Mask**

🟥 Edges & Shapes  🟨 Semantics

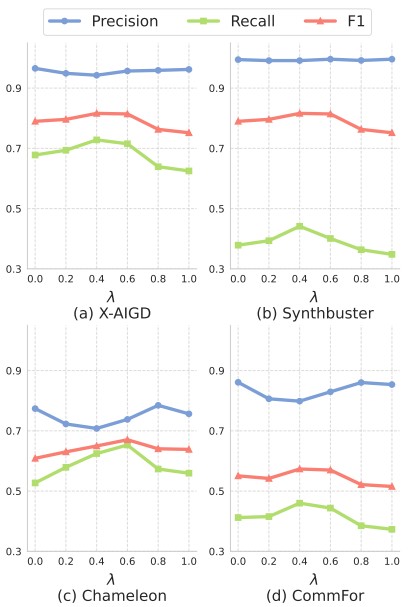

(a) X-AIGD    (b) Synthbuster

(c) Chameleon    (d) CommFor

Figure 6: **Comparison of interpretations of baseline and our artifact-aligned model.** Our model shows broader attention to potential artifact regions.

Figure 7: **Sensitivity analysis on benign region weight** $\lambda$. A moderate $\lambda$ (e.g., 0.4 or 0.6) yields higher $F_1$-scores.

to control the penalty on non-artifact features, leading to the following attention alignment loss:

$$\mathcal{L}_{align} = \frac{1}{hw} \sum_{i=1}^{h} \sum_{j=1}^{w} W^{(i,j)} \left( A_{cls}^{(i,j)} - A_{art}^{(i,j)} \right)^2, \quad \text{where} \quad W^{(i,j)} = \begin{cases} 1, & \text{if } A_{art}^{(i,j)} > 0 \\ \lambda, & \text{if } A_{art}^{(i,j)} = 0 \end{cases}. \quad (1)$$

By adding this regularization term to the standard binary cross-entropy loss $\mathcal{L}_{\text{BCE}}$ for authenticity judgment, the overall training loss becomes:

$$\mathcal{L} = \mathcal{L}_{\text{BCE}} + \beta \mathcal{L}_{align}, \quad (2)$$

where $\beta > 0$ is the weight for the attention alignment loss.

## 6.2 EXPERIMENTS

**Experiment setup.** To investigate whether explicit attention alignment can enhance the generalization of AIGI detectors, we finetune DINOv2 (Oquab et al., 2023) using the objective defined in (2) on the training set of X-AIGD, and evaluate the model on three AIGI detection datasets with diverse fake image sources in addition to the X-AIGD test set: Synthbuster (Bammey, 2023), Community Forensics (CommFor) (Park & Owens, 2025), and Chameleon (Yan et al., 2025). The hyperparameter selection is based on the accuracy on a validation set constructed from unannotated fake images generated by the same generators as the training set and the corresponding real images in X-AIGD. Additional details are provided in Appendix B.3.

**Ablation study.** We compare the artifact-based attention alignment method with a baseline that finetunes DINOv2 using only the binary cross-entropy loss (*i.e.*, $\beta = 0$). As shown in Tab. 4, the proposed method consistently improves the overall performance across datasets, especially on $F_1$-scores, demonstrating its effectiveness in enhancing model generalization. Furthermore, we conduct an additional ablation that replaces the annotated artifact masks with saliency masks generated by a pre-trained saliency detection model (Qin et al., 2020). The results indicate that regularizing model attention to salient objects does not yield similar performance improvements, underscoring the effectiveness of leveraging perceptual artifacts for attention alignment. Additional ablation studies are presented in Sec. D.4.

**Qualitative analysis.** Fig. 6 visualizes the interpretation heatmaps of the baseline and our artifact-aligned model. Our model shows significantly broader and more intensive attention to potential

artifact regions, such as the limbs of humans and animals, and complex structures or textures. This indicates that the our method effectively guides the model to focus on more interpretable features.

**Discussion on benign region weight** $\lambda$**.** A sensitivity analysis on $\lambda$ is illustrated in Fig. 7. A larger $\lambda$ discourages the model from utilizing detection clues beyond perceptual artifacts, such as low-level fingerprints or global characteristics. While perceptual artifacts can be more generalizable clues, the difficulty in detecting higher-level artifacts as suggested by Sec. 5 can lead to lower recall if the model overly relies on them. Therefore, balancing the contribution of perceptual artifacts and uninterpretable features is crucial for practical AIGI detection. In addition, the precision is consistently and significantly higher than the recall across different datasets, which relates to the common challenge identified by Ojha et al. (2023) that AIGI detectors often treat the real class as a *sink class* that hosts any sample not containing specific fake patterns. However, with moderate $\lambda$ values, the gap between precision and recall can be shrunk, indicating more balanced learning of both real and fake classes, and thus better $F_1$-scores in generalization.

## 7 CONCLUSION

To facilitate the research on interpretable and generalizable AI-generated image detection, we introduce X-AIGD, a fine-grained benchmark featuring pixel-level, categorized annotations of perceptual artifacts. X-AIGD enables fine-grained analysis of model decisions, uncovering critical insights into existing detection mechanisms. Our comprehensive evaluation reveals that existing AIGI detectors often overlook perceptual artifacts, relying instead on opaque, uninterpretable features. While multi-task learning shows potential for explicit artifact detection, it still struggles to align authenticity judgments with human-interpretable cues. Therefore, we propose an attention alignment method that regularizes model to focus on artifact regions, which significantly enhances interpretability and cross-dataset generalization. These findings highlight the significance of artifact-aware detection strategies. By setting a new standard for interpretability evaluation, X-AIGD aims to drive advancements in robust and explainable AIGI detection methods.

## ETHICS STATEMENT

**Research Involving Human Subjects or Participants**  To ensure the ethical conduct of this research, we conduct an internal ethics review to confirm compliance with relevant ethical standards. This study involves human participants recruited to perform data annotation tasks. Prior to participating, individuals are thoroughly informed about the purpose and significance of the research, the expected time commitment, and the details of compensation. Annotators are provided with flexible working hours, enabling them to complete tasks at their own convenience. To protect participant privacy, no personally identifiable information is collected beyond what is necessary for payment purposes. The identities of annotators are anonymized in all reporting and analysis. All personal payment-related information is handled confidentially and stored securely.

**Data-Related Concerns**  This study uses real-world image data from publicly available datasets, including the MSCOCO (Lin et al., 2014), the LAION-Aesthetic (LAION AI, 2022), the Conceptual Captions Dataset (Sharma et al., 2018), and the SA-1B (Kirillov et al., 2023). These datasets are curated by academic or non-profit organizations and are widely used for machine learning research under the appropriate licenses. We carefully review the licenses associated with each dataset to ensure compliance with usage restrictions and attribution requirements. Due to the nature of these large-scale web-crawled datasets, obtaining explicit consent from all potentially depicted individuals is not feasible. To mitigate potential ethical risks, we rely on expert models and human inspection to filter out NSFW (Not Safe For Work) content from both real and synthetic images. All usage is strictly for non-commercial academic purposes, aligning with the open science missions of the respective dataset creators. All synthetic images used in this study are generated from text prompts using open-source text-to-image models. The inputs are textual descriptions only, and the outputs are not derived from any specific real-world individuals or copyrighted works. Should any rights holder believe that their content has been used in violation of applicable copyright or privacy laws, we welcome them to contact us so that we may review and address the matter accordingly.

**Societal Impact and Potential Harmful Consequences**  This work introduces X-AIGD, a benchmark for interpretable AI-generated image detection, aiming to improve the transparency and reliability of identifying synthetic images. Although our goal is to enhance detection capabilities, we recognize

that such tools could be misused to reverse engineer and refine generation techniques, making AI-generated content even more realistic and harder to detect. This poses risks in terms of misinformation, deception, and societal manipulation. We emphasize the importance of responsible use and encourage researchers to consider the dual-use implications of this line of work.

**Impact Mitigation Measures** We provide a comprehensive dataset documentation, adopt an ethically responsible open license, ensure secure and privacy-preserving data practices, and enable full reproducibility by sharing all necessary research artifacts. We also confirm compliance with relevant legal and human rights standards. Our goal is to advance trustworthy, interpretable, and socially responsible research in AIGI detection.

## AUTHOR CONTRIBUTIONS

**Yao Xiao**: Conceptualization, Methodology, Software, Validation, Writing – Original Draft, Project Administration. **Weiyan Chen**: Conceptualization, Methodology, Software, Data Curation, Formal Analysis, Investigation, Visualization. Focus on the evaluation of existing models and the attention alignment method. **Jiahao Chen**: Conceptualization, Methodology, Software, Data Curation, Formal Analysis, Investigation. Focus on the investigation of MLLM-based interpretable AIGI detection, case study of text rendering artifacts, and annotation tool development. **Zijie Cao**: Conceptualization, Methodology, Software, Data Curation, Formal Analysis, Investigation, Visualization. Focus on multi-task models, qualitative result visualization, and annotation process monitoring. **Weijian Deng** and **Binbin Yang**: Writing – Review & Editing. **Ziyi Dong**: Conceptualization, Methodology. **Xiangyang Ji** and **Wei Ke**: Supervision. **Pengxu Wei**: Conceptualization, Writing – Review & Editing, Supervision, Project Administration. **Liang Lin**: Supervision.

## ACKNOWLEDGMENTS

This work is supported in part by National Natural Science Foundation of China (NSFC) under Grant No.62376292, 62376209, 62325605, 62536010, Guangdong Provincial General Fund No. 2024A1515010208, Guangzhou Science and Technology Program Project No.2025A04J5465, 2024A04J6365.

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

# A    ADDITIONAL DETAILS AND DISCUSSIONS OF X-AIGD BENCHMARK

## A.1    SOURCES OF REAL AND FAKE IMAGES

Tab. 5 summarize the real-image sources used to derive caption prompts for fake-image generation. The four sources provide 4,000 real images in total for the benchmark. Tab. 6 lists the 13 generators used to synthesize fake images, grouped by architecture. The checkmarks ✓ indicate which models contributed to the training and test splits used in our experiments.

Table 5: **Overview of real image sources in X-AIGD.**

| Dataset | Sample Count |
|---|---|
| SA-1B (Kirillov et al., 2023) | 2972 |
| Laion-Aesthetic (LAION AI, 2022) | 856 |
| MSCOCO (Lin et al., 2014) | 133 |
| Conceptual Captions Dataset (Sharma et al., 2018) | 39 |

Table 6: **Overview of image generation models used for fake image synthesis in X-AIGD.** These models cover three architectures: diffusion UNets (Rombach et al., 2022), Diffusion Transformers (DiT) (Peebles & Xie, 2022), and auto-regressive models. * indicates a community fine-tuned model on Civitai (Civ).

| Generator | Architecture | Training Set | Test set |
|---|---|---|---|
| SD v1.4 (Rombach et al., 2022) | Diffusion UNet | ✓ | ✓ |
| SD v1.5-relistic-vision* (SG_161222, 2024) | Diffusion UNet | ✓ | ✓ |
| SDXL (Podell et al., 2023) | Diffusion UNet | | ✓ |
| SDXL-realism-engine* (razzz, 2024) | Diffusion UNet | | ✓ |
| PixArt-$\alpha$ (Chen et al., 2024c) | DiT | ✓ | ✓ |
| Lumina-Next (Zhuo et al., 2024) | DiT | ✓ | ✓ |
| SD 3 (Esser et al., 2024) | DiT | ✓ | ✓ |
| HYDiT-v1.2 (Li et al., 2024b) | DiT | | ✓ |
| FLUX.1-dev (Labs, 2024) | DiT | | ✓ |
| FLUX.1-dev-iPhonePhoto* (Anibaaal, 2024a) | DiT | | ✓ |
| SD 3.5-L (Esser et al., 2024) | DiT | | ✓ |
| SD 3.5-L-iPhonePhoto* (Anibaaal, 2024b) | DiT | | ✓ |
| Infinity (Han et al., 2024) | Auto-regressive | | ✓ |

## A.2    IMAGE COLLECTION PROCESSES

> I provide an image and its corresponding content description text.
> Based on both the image and the provided description, please determine if the image was captured in a real scene (not generated by AI).
> If this photo was taken by a camera, respond with 'Yes'; otherwise, response with 'No'.
> Image content description text:

(a) Prompt template used for filtering unrealistic images from CC3M and Laion-Aesthetic.

> I provide an image and its corresponding content description text.
> Based on both the image and the provided description, please infer if the image has been blurred for privacy reasons (such as blurring of faces or license plates).
> If the image has been blurred, respond with 'Yes'. If the image has not been blurred, respond with 'No'.
> Image content description text:

(b) Prompt template used for filtering blurred images in SA-1B.

Figure 8: **Prompt templates used for real image filtering with InternVL2 8B (Chen et al., 2024d).**

**Real image filtering.** To ensure a diverse and realistic set of real-world images, we collect data from four widely-used datasets: COCO (Lin et al., 2014), CC3M (Sharma et al., 2018), Laion-Aesthetic (Schuhmann et al., 2022), and SA-1B (Kirillov et al., 2023). These datasets collectively

offer a broad variety of scenes, subjects, and visual complexities. To maintain a focus on authentic photographic content, we apply a filtering process to exclude non-photographic images. Specifically, we employ InternVL2 8B (Chen et al., 2024d) to identify and remove unrealistic images (e.g., cartoons and paintings) from CC3M and Laion-Aesthetic. Additionally, we exclude images from SA-1B that are blurred for privacy reasons. The detailed prompt templates are shown in Fig. 8.

**Text prompt collection.** We utilize captions that describe the visual content of real images as the text prompts for fake image generation, and these prompts are expected to span a wide range of complexity levels. To achieve this, we utilize BLIP-2 (Li et al., 2023) and InternVL2 8B (Chen et al., 2024d) to generate captions for real images. Notably, we employ different prompts to guide these models in producing captions of varying lengths, thereby controlling their descriptive richness and detail. In addition to model-generated captions, we further enrich our dataset by incorporating captions from COCO (Lin et al., 2014) and those used in (Chen et al., 2024c) for SA-1B. These sources, respectively, provide abundant examples of low- and high-complexity descriptions. By combining these sources and balancing the distribution of caption lengths, we construct a diverse and evenly distributed set of 4,000 captions, which serve as the text prompts for generating fake images.

**Prompt engineering for high-quality image generation.** To enhance image quality, we follow the prompt engineering approach in (Baraldi et al., 2024). Specifically, for the generation of each image, we randomly sample a set of positive modifiers and attach them to the text prompt. These modifiers include: `best quality, masterpiece, ultra highres, ultra realistic, HDR, photorealistic, hyper detailed, dynamic lighting,` and `professional lighting`. Each modifier is sampled independently with a probability of 0.5. In addition, we apply a fixed negative prompt to all generators: "`abstract, game, 3D style, fantastical, splash art, simple background, blurry background, blurry, Interchangeable Lens, Digital Photo Frame, rough, nsfw, lowres, bad anatomy, bad hands, text, error, missing fingers, extra digit, fewer digits, cropped, worst quality, low quality, normal quality, jpeg artifacts, signature, watermark, username, blurry`". These positive and negative modifiers apply to all the pre-trained generators. For the community fine-tuned models (razzz, 2024; Anibaaal, 2024a;b), which require specific modifiers to ensure the quality and style of generation, we follow the respective official recommendations.

**Safety checking.** For responsible data release, we adopt model-based image safety checking for the collected images. Specifically, we use an open-source NSFW image classifier (Falcons.ai, 2023) to filter out the unsafe images, and perform manual inspection on the positive results to eliminate the false alarms. Furthermore, during the annotation process, annotators are instructed to report any potentially harmful content in the generated images. A total of 12 generated images that do not pass the safety checks are excluded from the dataset.

### A.3 ANNOTATION OF PERCEPTUAL ARTIFACTS

**Image selection and train-test splitting for the annotated subset.** Given the labor-intensive nature of pixel-level annotation, we select a representative subset of the collected fake images to annotate. After the image collection process, we obtain 4,000 real images and 4,000 corresponding fake images from each of the 13 generators. We assign a unique UID for each real image and its corresponding fake images generated from the same prompt, resulting in 4,000 UIDs in total. We split half of the UIDs into the training set and the other half into the test set, ensuring no overlap between the two sets. Then, for both the training and test sets, we select 200 UIDs for each involved generator for annotation. This selection ensures that all generators have 100 overlapping UIDs in the training and test sets, while the remaining 100 UIDs are unique to each generator. This design allows for a comprehensive evaluation of cross-generator generalization and the impact of semantic diversity.

**Annotation workflow.** To reduce the impact of the unavoidable inter-annotator variation due to the subjective nature of artifact perception, our protocol is designed to prioritize *completeness* rather than force strict consensus among annotators. Instead of three fully independent passes, we employ a relay-style workflow: each annotator inspects the previous annotations, identifies missed artifacts, and adds any missing artifact regions when needed, including cases where their interpretation differs from earlier annotators. Each image undergoes three rounds of annotation by three different annotators.

Table 7: **Number of annotated artifact instances across all artifact types and generators.**

| Generator | Low-level | | | | High-level | Cognitive-level | | Total |
|---|---|---|---|---|---|---|---|---|
| | Textures | Edges&Shapes | Symbols | Color | Semantics | Commonsense | Physics | |
| FLUX.1-dev (Labs, 2024) | 27 | 520 | 143 | 16 | 182 | 50 | 38 | 976 |
| FLUX.1-dev-iPhonePhoto (Anibaaal, 2024a) | 85 | 599 | 201 | 36 | 105 | 19 | 20 | 1065 |
| HYDiT-v1.2 (Li et al., 2024b) | 78 | 618 | 38 | 20 | 271 | 23 | 9 | 1057 |
| Infinity (Han et al., 2024) | 50 | 747 | 82 | 25 | 272 | 49 | 24 | 1249 |
| Lumina-Next (Zhuo et al., 2024) | 155 | 1304 | 127 | 29 | 310 | 28 | 26 | 1979 |
| PixArt-$\alpha$ (Chen et al., 2024c) | 161 | 1200 | 131 | 30 | 358 | 27 | 40 | 1947 |
| SDXL (Podell et al., 2023) | 138 | 409 | 109 | 7 | 165 | 15 | 6 | 849 |
| SDXL-realism-engine (razzz, 2024) | 81 | 335 | 200 | 31 | 101 | 41 | 28 | 817 |
| SD v1.4 (Rombach et al., 2022) | 103 | 1189 | 94 | 35 | 286 | 24 | 18 | 1749 |
| SD v1.5-realistic-vision (SG_161222, 2024) | 285 | 1077 | 299 | 19 | 366 | 57 | 25 | 2128 |
| SD 3 (Esser et al., 2024) | 337 | 1220 | 327 | 28 | 463 | 79 | 44 | 2498 |
| SD 3.5-L (Esser et al., 2024) | 79 | 614 | 133 | 9 | 153 | 35 | 18 | 1041 |
| SD 3.5-L-iPhonePhoto (Anibaaal, 2024b) | 158 | 478 | 114 | 16 | 114 | 39 | 11 | 930 |
| Total | 1737 | 10310 | 1998 | 301 | 3146 | 486 | 307 | 18285 |

We intentionally retain all annotations rather than adjudicating them into a single canonical mask. A subsequent independent confidence-scoring step is employed to measure the annotator certainty across all annotations. Specifically, three independent annotators assign a confidence score of 0 (no artifact), 0.5 (ambiguous), or 1 (clear artifact) for each annotated artifact instance.

**Annotator instructions.** To ensure high-quality annotations, we enlist 12 annotators who possess a higher education background and foundational knowledge of AI-generated images. Before commencing annotation, all annotators undergo standardized training, which covers definitions of perceptual artifact categories and proficiency in annotation tool (*i.e.*, Labelme (Contributors)) operations. During the annotation process, annotators are required to meticulously inspect each generative image for perceptual artifacts, create precise polygon masks around the identified artifact regions, and categorize these artifacts into their correct classes according to standardized guidelines. Concurrently, annotators are instructed to flag out-of-scope images with unrealistic styles and unsafe images containing potentially harmful content.

**Annotation statistics.** Tab. 7 presents the statistics for annotated artifact instances within X-AIGD, encompassing both the training and test sets. A total of 18,285 artifact instances are annotated, spanning 13 generators and 7 distinct artifact categories (encompassing low-level, high-level, and cognitive-level artifacts).

### A.4 COMPARISON WITH RELATED ANNOTATED AIGI DATASETS

Tab. 8 provides a comprehensive comparison between X-AIGD and existing datasets with artifact annotations proposed for interpretable AIGI detection or perceptual artifact localization.

### A.5 EVALUATION METRICS FOR PERCEPTUAL ARTIFACT DETECTION

While this paper reports pixel-level PAD metrics (*i.e.*, IoU, PixP, PixR, and PixF$_1$) following prior works (Zhang et al., 2023; Kang et al., 2025), we argue that strict spatial precision is not necessarily required for interpretable AIGI detection, where approximate or partial localization is often sufficient for real-world applications.

To this end, we propose alternative instance-level matching metrics that tolerate partial localization. Specifically, a predicted region $\hat{r}_i$ with predicted category $\hat{c}_i$ is counted as a valid indication (*i.e.*, true positive) of a ground-truth artifact instance with region $r_j$ and category $c_j$ if $\hat{c}_i = c_j$ and $\text{Area}(\hat{r}_i \cap r_j)/\text{Area}(\hat{r}_i) \geq t$, where $t$ is a constant threshold. This parallels instance matching in object detection while accommodating the inherent fuzziness of artifact boundaries. For each artifact category, we compute the instance-level precision, recall, and F$_1$-score with threshold $t$, denoted by P@$t$, R@$t$, and F$_1$@$t$, respectively. These metrics are compatible with various forms of localized predictions, including pixel masks, bounding boxes, and points, thereby unifying the evaluation of models with different grounding abilities.

Table 8: **Comprehensive comparison with related annotated datasets.**

| Dataset | Real Images | Fake Images | | Artifact Annotation | | | |
|---|---|---|---|---|---|---|---|
| | | #Image | #Generator | Localization | #Category | #Instance | Annotator |
| FakeClue (Wen et al., 2025) | Paired | ~5,000 | Unknown | N/A | N/A | N/A | MLLM |
| FakeBench (Li et al., 2024a) | Unpaired | 3,000 | 10 | N/A | 14 | N/A | MLLM & Human |
| MMFR-Dataset (Gao et al., 2025) | Paired | 60,266 | Unknown | N/A | 5 | N/A | MLLM |
| FakeChain (Li et al., 2025) | Unpaired | 23,797 | 17 | N/A | 16 | N/A | MLLM & Human |
| PAL4VST (Zhang et al., 2023) | N/A | 10,168 | 11 | Pixel mask | N/A | 40,841 | Human |
| SynArtifact (Cao et al., 2024) | N/A | 1,310 | 8 | Bounding box | 13 | 1,593 | Human |
| RichHF-18K (Liang et al., 2024) | N/A | 11,140 | Unknown | Point | N/A | 82,430 | Human |
| LOKI (image) (Ye et al., 2025) | Unpaired | 229 | 10 | Bounding box | N/A | 687 | Human |
| SynthScars (Kang et al., 2025) | N/A | 12,236 | Unknown | Pixel mask | 3 | 26,566 | Human |
| X-AIGD (ours) | Paired | 3,337 (labeled) 48,651 (unlabeled) | 13 | Pixel mask | 7 | 18,285 | Human |

## A.6 MITIGATING THE IMPACT OF LABEL NOISE

Due to the inter-annotator variability of artifact annotations, training or assessing models with the annotated data without consideration of label noise can lead to model overfitting or less reliable evaluation results. To reduce the adverse effect of low-confidence annotations and fuzzy artifact boundaries on downstream use of X-AIGD, we recommend several complementary strategies: (1) exploiting the confidence scores for annotations: filter or re-weight the annotations according to their reliability; (2) training interpretable models with noise-tolerant localization loss; (3) evaluating model explanations via the instance-level weak-localization metrics proposed in Sec. A.5.

## B ADDITIONAL EXPERIMENTAL DETAILS

### B.1 ACCESSING INTERPRETATIONS OF AIGI DETECTORS

**Code and checkpoints for existing AIGI detectors.** We adopt the implementations and checkpoints for CNNSpot (Wang et al., 2020b), Gram-Net (Liu et al., 2020), and UnivFD (Ojha et al., 2023) from the AIGCDetectBenchmark (Zhong et al., 2024). For more recently proposed methods, including FatFormer (Liu et al., 2024b), DRCT-ConvB (Chen et al., 2024a), DRCT-CLIP (Chen et al., 2024a), and CoDE (Baraldi et al., 2024), we utilize the code and pre-trained models from their official repositories.

**Grad-CAM interpretation.** To evaluate the interpretability of CNN-like AIGI detectors, including Swin Transformers (Liu et al., 2021), we apply Grad-CAM (Selvaraju et al., 2017), as implemented in (Gildenblat & contributors, 2021), to generate heatmaps that highlight regions associated with the prediction of the "fake" class. In particular, we enable test-time augmentation smoothing (by setting `aug_smooth=True`) to obtain more stable and refined heatmaps; other parameters remain at their default settings. Furthermore, image pre-processing strategies dictate how full-image heatmaps are obtained. For resizing-based models (Wang et al., 2020b; Liu et al., 2020; Ojha et al., 2023; Liu et al., 2024b), heatmaps are directly mapped back to the entire image. In contrast, for patch-based models (Chen et al., 2024a; Baraldi et al., 2024), heatmaps are generated via a sliding-patch approach inspired by (Chai et al., 2020). Specifically, the model is applied to overlapping image patches (with window size and stride equal to 224), and the resulting per-patch heatmaps are then aggregated to form a full-image heatmap.

**Relevance Map interpretation.** Grad-CAM is not well-suited for Transformer models, as its methodology relies on the spatial locality inherent in CNNs, which is fundamentally violated by the global self-attention mechanism in Transformers. Therefore, we employ the Relevance Map method (Chefer et al., 2021), which is specifically designed for the Transformer architecture, to

visualize the attention regions of Transformer-based AIGI detectors (*i.e.*, FatFormer (Liu et al., 2024b), DRCT-CLIP (Chen et al., 2024a), CoDE (Baraldi et al., 2024), and the models studied in Sec. 6). All parameters are consistent with the official settings.

## B.2 MULTI-TASK MODEL TRAINING

**Multi-task model construction.** To explore the potential of learning PAD as an auxiliary task for interpretable AIGI detection, we construct a multi-task model that utilizes Swin Transformer (Liu et al., 2021) as the backbone. The same model architecture is applied to all variants studied in Sec. 5, including the single-task, transfer learning, and multi-task models. Specifically, we adopt the official Swin-B model pre-trained on ADE20K (Zhou et al., 2019). The features extracted by Swin-B are processed through two distinct branches: a binary classification head that consists of a global average pooling layer followed by a linear classifier with sigmoid activation (with the decision threshold fixed at 0.5), and a semantic segmentation head that employs UPerNet (Wang et al., 2020a) as the main head and is supplemented by an auxiliary FCN (Long et al., 2015) head during training. Both segmentation heads contain a hidden dimension of 512. During inference, the multi-task model simultaneously outputs the probability that the input image is fake and a pixel-level mask indicating various types of perceptual artifacts.

**Image augmentations.** To enhance the robustness of the multi-task model, we implement comprehensive image augmentation strategies during training. Given that fake images are typically in lossless PNG format while real images are predominantly JPEG-compressed, we apply random JPEG compression (sampling JPEG quality between 30 and 100) exclusively to fake images. This approach mitigates compression bias and better simulates real-world deployment conditions. For all images, we subsequently perform scale-preserving random resizing (with a resizing factor ranging from 0.5 to 2.0), random cropping (with a maximum cutout ratio of 0.75), and random flipping (with a 0.5 probability).

**Training details.** During training, the binary classification head for Authenticity Judgment (AJ) employs Binary Cross-Entropy loss. For the semantic segmentation heads addressing Perceptual Artifact Detection (PAD), we formulate this as a multi-class segmentation task, which considers one background class and the 7 classes of perceptual artifacts. For this subtask, we use a weighted Cross-Entropy loss where the class weights are approximately inversely proportional to their pixel-level frequencies in the training set. When combining the multi-task losses, the weights for the binary classification head, the main segmentation head, and the auxiliary segmentation head are set to 1.0, 1.0, and 0.2, respectively. To ensure class balance in AJ, we augment the labeled training sets with additional real images taken from the unlabeled training set (*i.e.*, those with UIDs that are split into the training set but not selected for annotation). We train the model with a batch size of 16 for 80,000 iterations. We utilize the AdamW optimizer (Loshchilov & Hutter, 2019) with a base learning rate of 2e-5 for the backbone and 2e-4 for the task heads. A small validation set is split from the training set for selecting the best checkpoint. Our code is built upon MMSegmentation (Contributors, 2020).

## B.3 ATTENTION ALIGNMENT

**Training details.** The training setup for the models in Sec. 6 generally follows that of the multi-task model in Sec. B.2, except that we use DINOv2 (Oquab et al., 2023) (ViT-B) as the pre-trained backbone and adopt a lighter fine-tuning recipe with less severe JPEG compression (60-100 quality factor) and no random resizing. The models are trained with a batch size of 8 for 20 epochs. We report the average results of 10 independent repeated experiments with different seeds to ensure the reliability of the results.

**Hyperparameter selection.** To determine the optimal value for the benign region weight $\lambda$, we create a validation set by randomly sampling 1,000 UIDs from the unannotated training set and ensuring that they have no overlap with the UIDs of images used for model training. The validation set shares the same 5 generators with the training set, and therefore has 1,000 real images and 5,000 fake images. We train the models with $\lambda \in \{0.0, 0.2, 0.4, 0.6, 0.8, 1.0\}$, and select the best $\lambda$ that yields the highest balanced accuracy on the validation set for each independent experiment. The weight of the attention alignment loss is fixed as 0.1 without heavy tuning.

Table 9: **Evaluation of MLLMs' ability to determine the presence of different types of artifacts in a fake image.** Image-level $F_1$-scores are reported for each artifact category.

| Model | Textures | Edges & Shapes | Symbols | Color | Semantics | Commonsense | Physics |
|---|---|---|---|---|---|---|---|
| Qwen2.5-VL 7B (Bai et al., 2025) | 5.0 | 5.3 | 49.0 | 14.6 | 26.7 | 11.7 | 14.3 |
| InternVL3 8B (Zhu et al., 2025) | 16.2 | 38.6 | 47.2 | 13.2 | 39.6 | 18.7 | 13.4 |
| LLaVa-Next (Mistral-7B) (Liu et al., 2024a) | 32.9 | 69.1 | 28.7 | 9.0 | 52.6 | 17.1 | 12.2 |
| DeepSeek-VL2-Small (Wu et al., 2024) | 32.3 | 71.1 | 33.2 | 7.8 | 45.2 | 18.5 | 17.1 |
| GPT-4o (OpenAI, 2024) | 9.3 | 59.5 | 91.7 | 0.0 | 40.5 | 32.3 | 9.1 |
| Random | 29.9 | 56.7 | 30.6 | 9.4 | 49.1 | 19.2 | 12.0 |

Table 10: **Authenticity judgment accuracy (%) of FakeVLM and FakeReasoning on X-AIGD.**

| Model | Real Acc. | Fake Acc. | Avg. Acc. |
|---|---|---|---|
| FakeVLM (Wen et al., 2025) | 43.7 | 98.9 | 71.3 |
| FakeReasoning (Gao et al., 2025) | 81.2 | 39.7 | 60.4 |

## C   DISCUSSIONS ON MLLM-BASED INTERPRETABLE AIGI DETECTION

A growing line of work (Gao et al., 2025; Li et al., 2025; Wen et al., 2025; Kang et al., 2025) attempts to build interpretable AIGI detection models based on MLLMs, leveraging their strong capacity of visual understanding and reasoning and the textual outputs for explanations. Some studies (Kang et al., 2025; Yang et al., 2025) integrate additional modules into MLLMs to enable grounded interpretations of image defects. In this section, we study the performance of existing MLLMs based on X-AIGD, and reveal the challenges of building MLLM-based interpretable AIGI detectors.

### C.1   EVALUATING PRE-TRAINED MLLMS ON PERCEPTUAL ARTIFACT DETECTION

To investigate whether pre-trained general-purpose MLLMs have sufficient ability to detect perceptual artifacts, we separately prompt an MLLM to identify each artifact category in a test fake image and return bounding boxes for detected instances, if any. We observe that most predicted boxes are highly imprecise, despite the models' general grounding capability on other high-level tasks. To obtain a meaningful measure, we evaluate only image-level binary predictions: any non-empty bounding-box list counts as predicting the presence of that artifact type. The image-level $F_1$-scores across the seven artifact categories are shown in Tab. 9. Across most categories, the MLLMs' predictions do not exceed random guessing, with the notable exception of the Symbols category. This suggests that current MLLMs may lack the fine-grained perceptual sensitivity required for grounded artifact reasoning, including the proprietary model GPT-4o.

### C.2   EVALUATING MLLM-BASED INTERPRETABLE AIGI DETECTORS

We conduct a focused evaluation of three open-source representatives of this line of work, namely FakeVLM (Wen et al., 2025), FakeReasoning (Gao et al., 2025), and LEGION (Kang et al., 2025), using the X-AIGD benchmark.

**Authenticity judgment performance.** The textual output of FakeVLM and FakeReasoning comprises a binary authenticity prediction and accompanying explanations. When evaluated on X-AIGD, both models exhibit significant class bias, resulting in relatively low average accuracy, as shown in Tab. 10. In addition, the LEGION model trained for artifact localization and explanation produces a non-zero artifact mask and at least one artifact description for *every* input image, including real ones. This behavior can be explained by the fact that this model is trained exclusively on fake images and never observes real image samples.

**Qualitative assessment of explanations.** As exemplified by Figs. 9 and 10, our qualitative analysis of the explanations produced by the three models highlights several recurring issues:

- FakeVLM and FakeReasoning: Often present vague or ungrounded judgments such as "inconsistent light source", "lacks realistic detail", or "(objects) do not match reality". While some common cues (e.g., "overly smooth", "saturation is too high") are correct in some cases, they frequently appear as false positives. These patterns likely stem from the MLLMs used for generating the annotations for their training data.

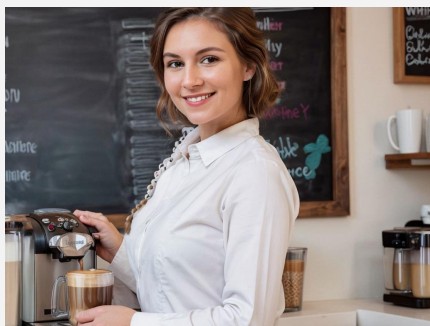

Authenticity: **fake**

**FakeVLM: fake**
**Explanation**: This is a **fake** image. Although it does not show obvious artifacts or direct signs of composition, **there are underlying characteristic inconsistencies in its features that suggest it is artificially created.**

**FakeReasoning: real**
**Explanation**: <SUMMARY> I will determine if the image is real or fake by analyzing its content and identifying forgery attributes from both low-level and high-level perspectives. I will then conclude by stating either 'this image is real' or 'this image is fake' based on my analysis. </SUMMARY>
<CAPTION> The image shows a woman standing behind a coffee bar, smiling and holding a cup of coffee. She is wearing a white blouse and appears to be in a cozy, rustic setting with chalkboard menus on the wall listing various drink options. The background includes coffee machines and other kitchen items, suggesting a café or home kitchen environment. </CAPTION>
<REASONING> **From low-level perspective:** The textures of the person's clothing, the wooden background, and the coffee equipment **appear natural and detailed.The edges of objects**, such as the person and the coffee machine, **The lighting on the person and the shadows cast by** objects are consistent with a natural light source.The image is clear and sharp, with good resolution and color balance. **From high-level perspective:** The scene depicts a person in a coffee shop setting, which is a common and realistic scenario. Therefore, this image is real. </REASONING>
<CONCLUSION> **This image is real.** </CONCLUSION>

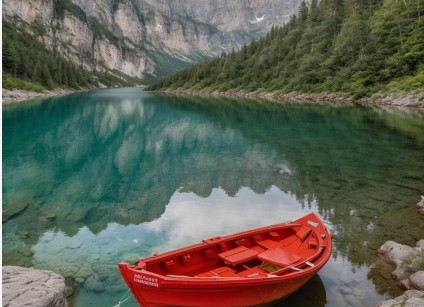

Authenticity: **fake**

**FakeVLM: fake**
**Explanation**: This is a fake image. **The image exhibits underlying characteristic inconsistencies in its features that suggest it is artificially created.**

**FakeReasoning: fake**
**Explanation**: <SUMMARY> I will determine if the image is real or fake by analyzing its content and identifying forgery attributes from both low-level and high-level perspectives. I will then conclude by stating either 'this image is real' or 'this image is fake' based on my analysis. </SUMMARY>
<CAPTION> The image shows a serene lakeside scene with a vibrant red rowboat resting on the shore. The boat is positioned near a rocky area, and the calm, clear water reflects the surrounding greenery and mountains in the background. The sky above is partly cloudy, adding to the tranquil atmosphere of the setting. </CAPTION>
<REASONING> **From low-level perspective: The lighting** on the boat does not match the lighting on the surrounding environment, suggesting inconsistency. **The texture of the water and the boat appears overly smooth and lacks realistic detail. From high-level perspective: The boat appears to be floating on the water without any visible means of support, which is unusual**. Therefore, this image is fake. </REASONING>
<CONCLUSION> This image is fake. </CONCLUSION>

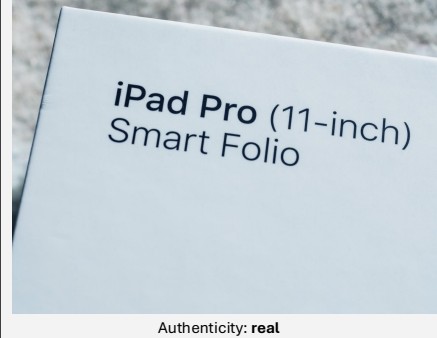

Authenticity: **real**

**FakeVLM: fake**
**Explanation**: This is a fake image. **The box has an unrealistic shape and structure**, with an overly smooth and uniform surface that **lacks natural imperfections**. The edges and corners are too perfect and lack the slight irregularities typically found in real-world objects. **The lighting and shadows are inconsistent**, with an unnatural glow that does not align with the expected light source. **The overall appearance is too perfect and lacks the subtle variations and imperfections** that would be present in a real object.

**FakeReasoning: real**
**Explanation**: <SUMMARY> I will determine if the image is real or fake by analyzing its content and identifying forgery attributes from both low-level and high-level perspectives. I will then conclude by stating either 'this image is real' or 'this image is fake' based on my analysis. </SUMMARY>
<CAPTION> The image shows a white box labeled \"iPad Pro 11-inch Smart Folio.\" The text is in bold, black letters against a light background, indicating it is a product packaging for an iPad Pro case. The box appears to be new and unopened, with a clean, minimalist design. </CAPTION>
<REASONING> **From low-level perspective: The texture of** the box and the surface it is on appears natural and consistent with real-world materials.**The edges of** the box and the text are smooth and well-defined, typical of a real photograph.**The lighting and shadows on** the box are consistent with a natural light source.The image is clear and detailed, indicating it is a high-quality photograph. **From high-level perspective**: The text on the box is coherent and logical, indicating a real product packaging. Therefore, this image is real. </REASONING>
<CONCLUSION> This image is real. </CONCLUSION>

**Observations:**
**1. FakeVLM appears to show a tendency towards the "fake" class in these challenging cases.** In these observed instances, its justifications resort to **generic templates** rather than specific visual evidence. For instance, it may use broad descriptions like "inconsistencies in features" without explicitly grounding them in the image content.
**2.** In contrast, **FakeReasoning** is **overly conservative**, favoring "real" predictions. The provided explanations might discuss valid image features (like shadows) but **miss the actual artifacts**, making the reasoning less specific to the defects.

Figure 9: **Qualitative examples of the judgments and explanations of FakeVLM and FakeReasoning.** Images are selected from X-AIGD.

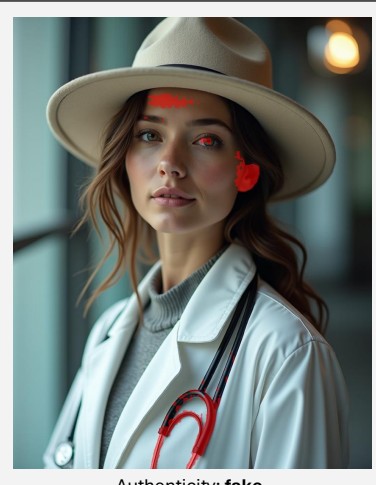

**LEGION**
**Explanation**: Upon examining the image. I have found: A woman in a white coat and hat stands confidently, with her left eye's tail tilted to the lower left and the pupil enlarged, her right eye's tail tilted to the upper left and the pupil enlarged, her left ear's outer rim, ear bone, and earlobe fused, her right ear's outer rim, ear bone, and earlobe fused, and her left hand's fingers twisted and abnormally shaped.
To elaborate, I have found the following artifacts.
Lady's left eye :The tail of the left eye is tilted to the lower left, the pupil is enlarged, and the appearance is unnatural.
Lady's right eye :The tail of the right eye is tilted to the upper left, the pupil is enlarged, and the appearance is unnatural.
Lady's left ear :The outer ear rim, ear bone and earlobe of the left ear are fused together.
Lady's right ear :The outer ear rim, ear bone and earlobe of the right ear are fused together.
Lady's left hand :The fingers of the left hand are twisted and abnormally shaped, which is obviously different from normal fingers.

Authenticity: **fake**

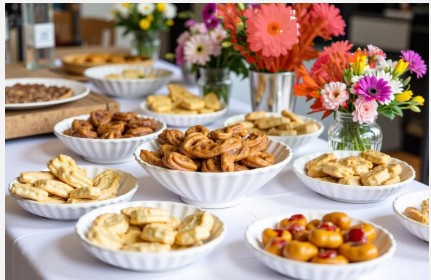

**LEGION**
**Explanation**: Upon examining the image. I have found: A beautifully arranged table features a variety of delectable treats, including cookies, fruit, and bread, with a vibrant bouquet of flowers in the background.
To elaborate, I have found the following artifacts.
Flowers in the vase on the right side of the table :The flowers in the vase on the right side of the table are not the same as the flowers in the vase on the left side of the table.

Authenticity: **fake**

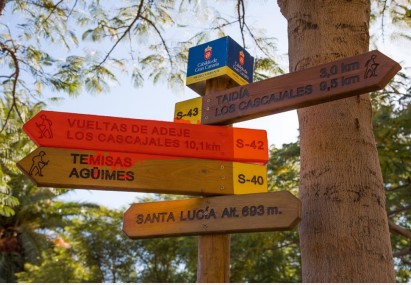

**LEGION**
**Explanation**: Upon examining the image. I have found: A wooden signpost with a colorful box on top stands in a sunlit outdoor setting, with the text on the signs below appearing distorted and illegible.
To elaborate, I have found the following artifacts.
Text below the signs :The text on the signs below is distorted and illegible.

Authenticity: **real**

**Observations:**
**In the demonstrated examples, LEGION appears to exhibit signs of hallucination.** It generates masks and explanations for real images where no obvious defects exist. Furthermore, **these visual masks do not always align with the accompanying text**. Finally, the model seems to exhibit a **hyper-sensitivity towards text regions**: even legible text is sometimes misclassified as "distorted" or "indecipherable".

Figure 10: **Qualitative examples of the judgments and explanations of LEGION.** Images are selected from X-AIGD. The predicted artifact masks are marked on the images with red masks.

- LEGION: While its binary masks can localize some artifact regions, the generated textual explanations may be inconsistent with the predicted masks. We observe hallucinated object parts (e.g., nonexistent hands) or descriptions of abnormalities that do not correspond to any detected region, suggesting potential overfitting to textual patterns rather than grounded visual evidence.

**Quantitative evaluation of artifact localization for LEGION.** We evaluate the artifact masks produced by LEGION on the Category-Agnostic PAD task, and the results are shown in Tab. 11. Compared with the interpretations of the DINOv2 models trained in Sec. 6, LEGION outperforms a vanilla fine-tuned baseline in all metrics but remains inferior to our attention-alignment model trained on our annotated data.

Table 11: **Quantitative comparison of LEGION's artifact masks and our model interpretations on *Category-Agnostic PAD*.**

| Model | IoU | PixP | PixR | PixF$_1$ |
|---|---|---|---|---|
| Baseline | 0.5 | 24.4 | 0.5 | 1.0 |
| LEGION (Kang et al., 2025) | 9.1 | 29.1 | 11.7 | 16.7 |
| Ours (Attention Alignment) | 14.0 | 37.2 | 18.2 | 24.5 |

**Summary.** These findings highlight the importance of X-AIGD. Existing MLLM-based methods may generate explanations that are ungrounded, inconsistent, or biased, and they lack access to paired real and generated images with fine-grained human annotations that are essential for reliable interpretability evaluation. X-AIGD addresses these limitations by offering localized and trustworthy supervision, which enables principled analysis of whether models truly attend to perceptual artifacts rather than spurious cues.

### C.3 IMPROVING MLLMS ON PERCEPTUAL ARTIFACT DETECTION

To investigate whether our annotated data can be exploited to improve MLLMs' ability to detect perceptual artifact, we perform a preliminary experiment on Qwen2.5-VL 7B (Bai et al., 2025) using GRPO-based reinforcement fine-tuning (Shao et al., 2024; Liu et al., 2025). Specifically, in each rollout, the model is prompted to predict the authenticity of the input image and provides a list of detected artifact instances that are localized by bounding boxes and categorized following the taxonomy of X-AIGD. We adopt the classification reward, IoU-based detection reward, and format reward following Visual-RFT (Liu et al., 2025).

To assess the PAD performance of the models, we adopt the instance-level metric discussed in Sec. A.5. The F$_1$@0.5 scores for the pre-trained Qwen2.5-VL 7B and our fine-tuned model are presented in Tab. 12. First, the pre-trained model shows weak initial competence on most artifact types except for the Symbols catagory, aligning with our findings in Sec. C.1. As a result, the effectiveness of reinforcement fine-tuning is limited: it yields notable gains for the Symbols category, while the improvements for other artifact types are at most marginal. Overall, these observations indicate that X-AIGD can improve the perceptual artifact detection ability of MLLMs on certain artifact categories, although broader gains likely require more effective training strategies and complementary supervision such as pseudo-labeled synthetic data as discussed in Sec. E.

## D    MORE VISUALIZATIONS AND EXPERIMENTAL RESULTS

### D.1    VISUALIZATION OF ANNOTATED ARTIFACTS

In Fig. 1, we visualize the annotated artifacts for each category. We provide more examples in Fig. 11 to further demonstrate the artifact categories and our pixel-level annotations.

### D.2    COMPARISON OF THE MODEL INTERPRETATIONS

In Fig. 4, we compare the interpretation heatmaps of different models and the segmentation results of the multi-task model against the ground truth artifact regions. We provide more visualization results in Fig. 12.

### D.3    QUANTITATIVE EVALUATION ON MODEL INTERPRETATIONS.

To quantitatively study the alignment between model interpretations and human-annotated artifact regions for the models in Sec. 5 and Sec. 6, and validate the effectiveness of our attention alignment method, we compare the Category-Agnostic PAD performance of these models as in Sec. 4. As shown

Table 12: **PAD performance of Qwen2.5-VL 7B before and after GRPO fine-tuning.** Instance-level $F_1$-scores at threshold $t = 0.5$ (as defined in Sec. A.5) are reported for each artifact category.

| Model | Textures | Edges & Shapes | Symbols | Color | Semantics | Commonsense | Physics |
|---|---|---|---|---|---|---|---|
| Qwen2.5-VL 7B (Bai et al., 2025) | 1.3 | 0.3 | 18.8 | 3.0 | 1.3 | 0.0 | 0.0 |
| Fine-tuned | 1.9 | 1.2 | 31.5 | 1.9 | 3.2 | 2.5 | 2.5 |

Table 13: **Quantitative evaluation on model interpretations on *Category-Agnostic PAD*.** The Swin-based models are studied in Sec. 5, while the DINOv2-based models are trained in Sec. 6.

| Backbone | Method | Category-Agnostic PAD | | | |
|---|---|---|---|---|---|
| | | IoU | PixP | PixR | PixF1 |
| Swin | AJ-only | 8.8 | 11.8 | 25.9 | 16.2 |
| | Transfer learning (linear probing) | 6.3 | 10.1 | 14.2 | 11.8 |
| | Transfer learning (full fine-tuning) | 6.9 | 15.2 | 11.3 | 13.0 |
| | Multi-task learning | 4.8 | 11.0 | 7.8 | 9.2 |
| DINOv2 | Baseline | 0.5 | 24.4 | 0.5 | 1.0 |
| | Attention Alignment | 14.0 | 37.2 | 18.2 | 24.5 |

in Tab. 13, the interpretations of our artifact-based attention alignment model achieve significantly better alignment with the annotated artifacts, while the transfer learning and multi-task models studied in Sec. 5 achieve no improvement compared with the single-task model. These observations are in line with our qualitative results in Sec. 5 and Sec. 6.

## D.4 Additional Ablations and LoRA Fine-Tuning for Attention Alignment

The experiments in Sec. 6 are conducted based on full parameter fine-tuning on DINOv2 with a small learning rate of 1e-5. Our observations on the training process suggest that all variants in the ablation study tend to rapidly collapse to similar optima, limiting the observable differences between training configurations. In addition, the Gradient Attention Rollout (Gildenblat, 2020) used for aggregating the model attention requires backward passes to obtain attention gradients, complicating the computation graph and preventing gradient accumulation. This increases training noise and reduces the stability of the comparisons.

To further validate the effectiveness of attention alignment, we experiment with a more stable and scalable training recipe, applying LoRA to fine-tune the DINOv2 backbone with rank=8 and learning rate of 1e-4 with cosine annealing schedule (Loshchilov & Hutter, 2017). In addition, we use the standard Attention Rollout (Abnar & Zuidema, 2020) for attention aggregation in this setup, which does not require computing attention gradients and allow us to achieve an effective batch size of 32.

With this training recipe, we conduct additional ablation of the proposed artifact-based attention alignment with two types of masks: (i) **Random** mask: a randomly sampled convex region within the image; (ii) **Benign** mask: the largest SAM-segmented region that contains no overlap with any annotated artifact mask (to avoid noisy small fragments). We train the models using these masks under the same attention alignment objective and compare them with the proposed artifact-based alignment. The benign-region weight $\lambda$ and the alignment-loss weight $\beta$ are set to 0.4 and 1.0 for all variants without further tuning. The results in Tab. 14 suggest that both Benign and Random masks yield results comparable to or worse than the no-alignment baseline, whereas artifact-based alignment consistently provides the largest improvements across datasets. These findings indicate that the performance gains indeed stem from aligning attention with perceptual artifact regions rather than generic spatial regularization effects.

Table 14: **Additional ablation study on attention alignment.** Artifact-based alignment outperforms the alignment with other types of masks (Benign, Random, and Saliency) across datasets.

| Alignment | X-AIGD | | Synthbuster | | Chameleon | | CommFor | |
|---|---|---|---|---|---|---|---|---|
| Mask | Acc | $F_1$ | Acc | $F_1$ | Acc | $F_1$ | Acc | $F_1$ |
| None ($\beta = 0$) | 82.2 | 82.9 | 71.2 | 60.6 | 68.0 | 68.2 | 72.1 | 63.6 |
| Benign | 80.7 | 81.4 | 70.6 | 60.3 | 65.6 | 66.4 | 71.5 | 63.1 |
| Random | 81.6 | 82.3 | 69.7 | 57.6 | 66.6 | 67.3 | 72.3 | 64.1 |
| Saliency | 81.3 | 81.9 | 71.4 | 61.8 | 66.0 | 66.5 | 72.4 | 64.5 |
| Artifact | **83.1** | **84.0** | **71.8** | **62.3** | **68.7** | **68.8** | **73.7** | **66.5** |

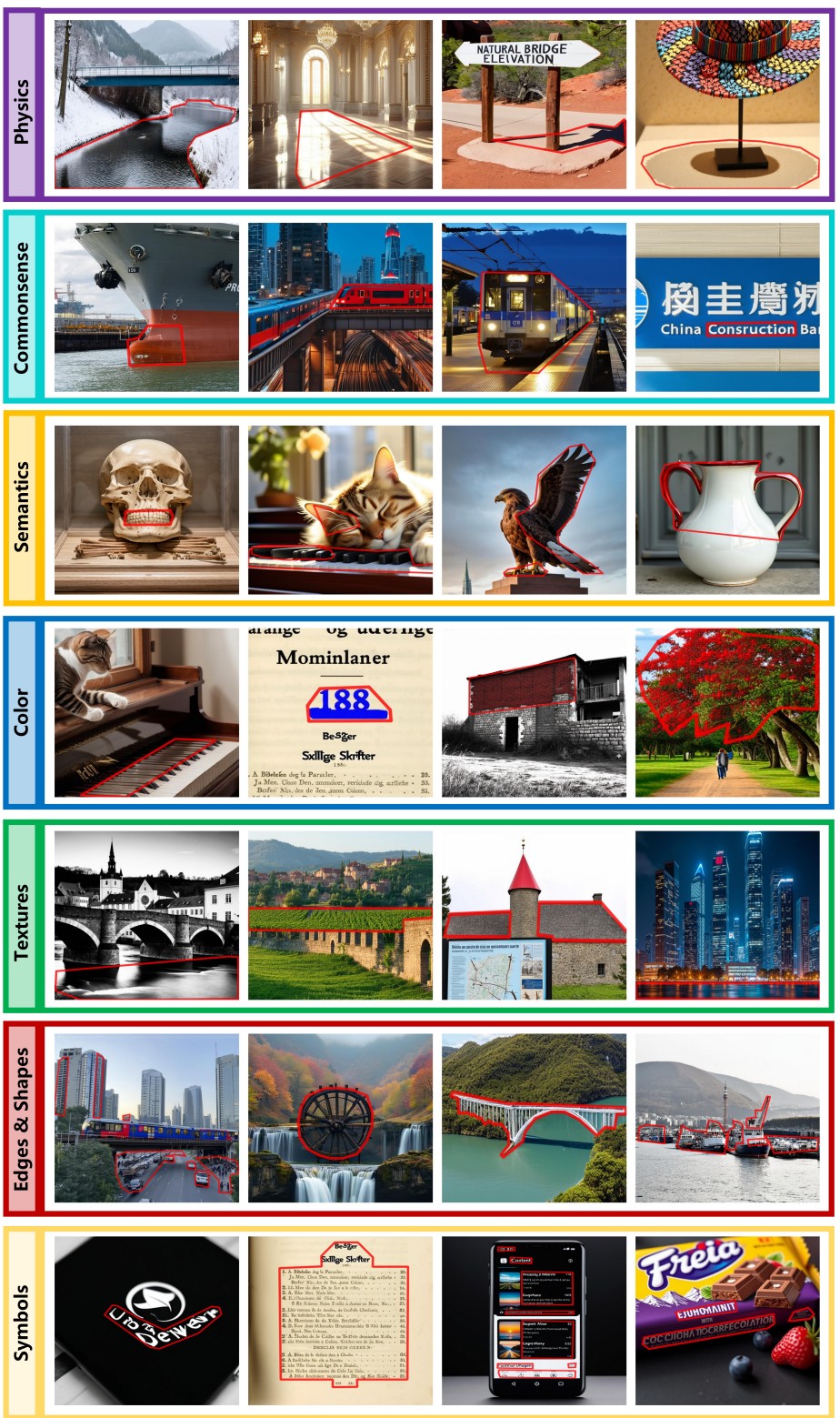

Figure 11: **Visualization of annotated perceptual artifacts.**

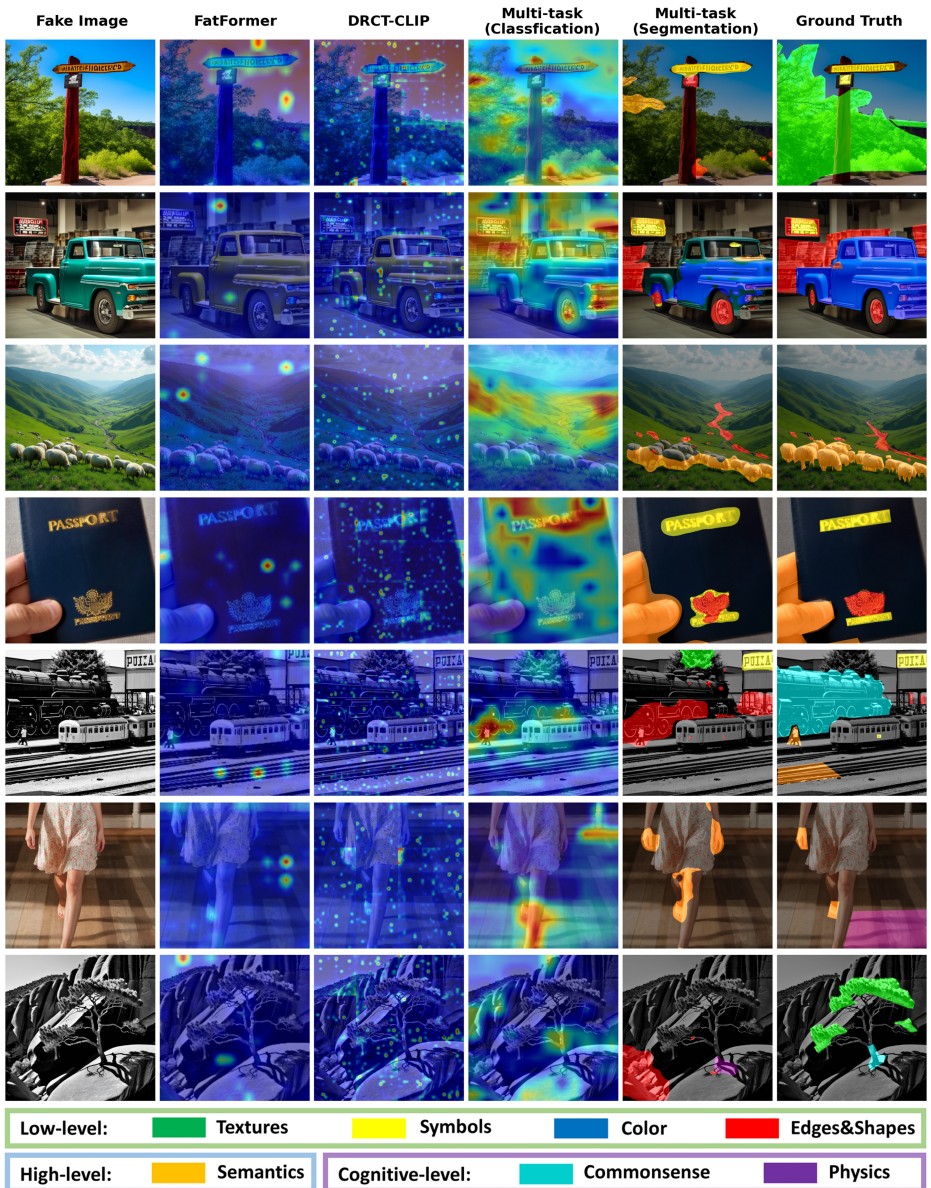

Figure 12: **Results of the interpretations of different models.**

# E    LIMITATION AND FUTURE DIRECTION

Our benchmark provides pixel-level, categorized annotations of perceptual artifacts, enabling fine-grained analysis of AI-generated image detection. While our evaluation primarily focuses on Authenticity Judgment and Perceptual Artifact Detection, future directions include expanding to multi-dimensional analyses, such as human alignment, image realism, and diversity metrics, to further enhance interpretability and robustness in detection systems. To analyze the potentials and challenges of artifact-aware interpretable AIGI detection, we provide several simple baselines such as multi-task learning and attention alignment, shedding light on its inherent complexity. Future work could explore advanced methodology that better utilizes perceptual artifact for enhanced interpretability and performance. Additionally, while the empirical studies in this paper focus on traditional classification and segmentation models, future research could explore grounded evaluation of Multimodal Large Language Models (MLLMs) and their capabilities in boosting artifact-aware AIGI detection, which requires more sophisticated techniques and represents an intriguing direction.

Moreover, training MLLMs for this purpose may require scaling up the annotated data, but it is intractable to rely exclusively on human annotation due to the high cost. Promising directions for cost-effective annotated data construction include (1) semi-automatic expansion based on models trained on X-AIGD and fine-tuned with the human-in-the-loop paradigm, and (2) using controlled image editing to produce pseudo-labeled fake images where the automatically generated editing masks serve as a coarse localization of potential artifact regions.

## F  BROADER IMPACT

With the continuous advancement and widespread diffusion of AIGI, it has become increasingly challenging for humans to distinguish synthetic images from real images. This poses a significant risk, as malicious actors can exploit synthetic images for creating misinformation, defamation, and fraud, thereby eroding public trust in digital media. To counter these threats, researchers have developed AIGI detectors. However, a critical limitation of current detectors is their lack of explainability; they typically provide only a binary classification (fake or real) without offering insight into why a particular image is flagged.

Addressing this gap, our work introduces X-AIGD, a novel benchmark specifically designed to promote the development and evaluation of explainable synthetic image detection methods. X-AIGD incorporates detailed annotations of human-perceptible artifact regions within synthetic images. Our analysis, utilizing this benchmark data, reveals a significant discrepancy: current AIGI detectors often focus on image regions that do not align with artifacts easily perceivable by humans, making their decision-making process opaque and untrustworthy from a human perspective. This finding underscores the urgent need for detectors that are not only accurate but also provide understandable explanations for their judgments.

With the proliferation of AIGI, high-performance AIGI detectors are crucial for news organizations, social media platforms, fact-checking organizations, and the general public to identify and resist misinformation, defamation, fraud, and other issues. Furthermore, providing evidence (explainability) alongside the detection results can further enhance persuasiveness. By annotating the artifact regions in generated images, we point out the limitations that generation techniques have during the generation process. This might potentially stimulate further development of generation techniques, which, while aimed at improvement, could also inadvertently make generated content harder to detect in the future, potentially exacerbating misinformation, deception, and societal manipulation if detection methods do not keep pace.

Our work aims to contribute to building a more transparent and trustworthy digital information environment. By providing a fine-grained benchmark and in-depth analysis, we hope to stimulate more research on the explainability and reliability of AIGI detection, ultimately helping society better navigate the opportunities and challenges brought about by AI technology.

## G  LARGE LANGUAGE MODEL (LLM) USAGE

We used Large Language Models (LLMs) as a general-purpose assistant for writing and editing this paper. Specifically, LLMs were used for: (1) text refinement: to aid in refining and polishing the manuscript, including improving clarity, grammar, and sentence structure; (2) experiment script generation and editing: to assist in writing and debugging some of the experiment scripts used for data processing, results aggregation, *etc*.

LLMs were not used for research ideation, methodology, or analysis. All core ideas, research contributions, and experimental results presented in this paper are the original work of the authors. The authors have reviewed all LLM-generated content and take full responsibility for the final contents of this paper.

