# OpenReview forum: "Unveiling Perceptual Artifacts: A Fine-Grained Benchmark for Interpretable AI-Generated Image Detection"
_ICLR.cc/2026/Conference — ICLR 2026 Poster_

### Official Review · Reviewer_NFjz · 2025-10-17

**Soundness:** 3
**Presentation:** 3
**Contribution:** 3
**Rating:** 4
**Confidence:** 5

**Summary:**

The paper introduces X-AIGD, a benchmark for interpretable AI-generated image (AIGI) detection featuring pixel-level masks and a three-level taxonomy of perceptual artifacts (low-level distortions, high-level semantics, cognitive-level counterfactuals). Data are paired: each fake is generated from a caption of a real image using 13 modern generators; a subset is densely annotated via multi-round human labeling. Two tasks are specified: Authenticity Judgment (AJ) and Perceptual Artifact Detection (PAD) (category-aware and category-agnostic). Empirically, the authors show: (i) popular detectors largely do not align with human-perceived artifacts; (ii) PAD as an auxiliary task only marginally improves AJ; (iii) an attention-alignment loss that encourages ViT attention to overlap artifact masks,

**Strengths:**

1. The paper tackles a core need in AIGI detection: explainability grounded in evidence. X-AIGD’s large, human-effort annotation effort is itself a significant contribution—creating supervision that most existing benchmarks lack and that practitioners actually need to diagnose model behavior.


2. Thoughtful three-tier taxonomy (low → semantic → cognitive): The taxonomy organizes artifacts by perceptual level, which is both conceptually coherent and practically useful. It supports analyses across levels (e.g., low-level texture vs. semantic object integrity vs. cognitive/commonsense violations), and invites targeted model improvements rather than one-size-fits-all heuristics.

**Weaknesses:**

1. My own experience annotating AIGC artifacts suggests substantial inter-annotator variation—especially for semantic flaws and low-level distortion— that the paper does not adequately discuss this important issue.

   * **Granularity ambiguity.** For the “wrong number of fingers” example in Fig. 1, some annotators box the entire palm while others isolate only the finger region. For the misspelled “McDonald’s,” some include the apostrophe while others do not. Both are defensible, but they produce inconsistent spatial supports and labels that are hard to reconcile.
   * **Diffuse low-level cues.** Low-level distortions (e.g., “over-smooth skin” to annotate a fake person) are inherently fuzzy; deciding “how much” of a face to include varies widely by annotator.
   * **Why it matters:** This variance directly affects the reliability of the proposed benchmark and any conclusions drawn from artifact-aligned training.

2. Insufficient design for precise and unified human annotation.  The value of X-AIGD hinges on label precision and consistency, yet the paper offers little methodology to *achieve* them.

   * **Missing protocol depth.** There is no clear calibration phase, no adjudication workflow for contentious cases, and no rubric that operationalizes the taxonomy (positive/negative exemplars, near-miss guidance).
   * **Coarse quality assessment.** Fig. 2 provides only a coarse human assessment; there is no formal inter-annotator reliability.
   * Without rigorous agreement analysis and a documented process to reduce disagreement, it is hard to trust the benchmark as a gold standard rather than a noisy training signal.

3. Marginal ablation gains for attention alignment (Table 4).  The reported improvements are small and could plausibly be explained by training noise or seed variance.

**Questions:**

1. Comprehensive assessment of X-AIGD: expand the dataset validation beyond the current coarse plots and include a side-by-side comparison with other annotated resources (e.g., Legion) along these dimensions.

2. Synthetic edit–driven annotations as a unification strategy: Human annotations of semantic/low-level flaws are noisy. Why not also leverage modern image-edit models (e.g., FLUX-kontext, GPT-4o) to produce programmatically controlled flaws with auto-derived pixel masks from edit operations? Please discuss pros/cons and provide an experiment comparing X-AIGD (human) vs Edit-AIGD (synthetic edits) vs Hybrid (human+synthetic).

3. Using X-AIGD to strengthen VLMs under fair conditions: Since X-AIGD offers pixel-level, typed supervision, it’s natural to test whether it measurably improves a strong VLM (e.g., Qwen2.5-VL) on AJ and PAD. Add a controlled finetune where only supervision differs (with/without artifact masks), holding training budget, data volume, and augmentations constant across VLM baselines.

---

> ### Author Response · Authors · 2025-11-27
>
> **W1 & W2: Inter-annotator variation and methods to handle inconsistency are not adequately discussed.**
>
> Thank you for these insightful comments. We acknowledge that perceptual artifact annotation could involve nuanced boundary decisions, and we clarify our quality-control strategy below.
>
>  **(1) Clarification on our annotation workflow & reducation of inter-annotator variation.**
> Our preliminary annotation trials also revealed substantial inter-annotator variation. To reduce its impact, our protocol is designed to prioritize completeness rather than force strict consensus. Instead of three fully independent passes, we employ a relay-style workflow: each annotator inspects the previous annotations, identifies missed artifacts, and adds any missing artifact regions when needed, including cases where their interpretation differs from earlier annotators. We intentionally retain all annotations rather than adjudicating them into a single canonical mask. A subsequent independent confidence-scoring step (summarized in Fig. 2) provides a measure of annotator certainty across these labels.
>
> **(2) Mitigating the impact of label noise.**
> To reduce the effect of inter-annotator variability on downstream use, we adopt two complementary strategies:
> - **Releasing full confidence scores for all annotations.** In the final revision, we will extend confidence scoring to all annotations, and release per-annotator confidence scores as part of the metadata. This enables future users to filter, reweight, or threshold annotations according to their reliability.
> - **Noise-tolerant evaluation via instance-level weak-localization metrics.**
> While the paper reports pixel-level PAD metrics following prior work, we believe that strict spatial precision is not necessarily required for interpretable AIGI detection, where approximate or partial localization is often sufficient for real-world applications. To address boundary ambiguity and subjective granularity, we additionally employ instance-level matching metrics that tolerate partial localization. Specifically, a predicted region $\hat{r}_i$ with predicted category $\hat{c}_i$ is counted as a valid indication of a ground-truth artifact instance with region $r_j$ and category $c_j$ if $\hat{c}_i=c_j$ and $\text{Area}(\hat{r}_i \cap r_j) / \text{Area}(\hat{r}_i) \geq t$, where $t$ is a constant threshold. This parallels instance matching in object detection while accommodating the inherent fuzziness of artifact boundaries. It is also better suited for evaluating MLLMs' artifact grounding abilities.
>
> The clarification on our annotation workflow has been updated in **Appendix A.3** of the revision, and the discussions on mitigating the impact of inter-annotator variance are included in **Appendix A.6**. The instance-level metrics for PAD are introduced in **Appendix A.5**.

---

> ### Author Response · Authors · 2025-11-27
>
> **W3: Marginal ablation gains for attention alignment (Table 4) ... plausibly be explained by training noise or seed variance.**
>
> Thank you for the comment. While the improvements in Table 4 are small, they consistently appear across datasets and seeds. Combined with the qualitative attention visualizations in Fig. 6, the results indicate that the effect is stable rather than attributable to training noise.
>
> To further validate the effectiveness of attention alignment, we introduce an additional baseline that applies LoRA to fine-tune DINOV2 instead of full-parameter training, which stabilizes optimization and reduces variance. The updated results based on *DINOV2-LoRA* are reported below:
>
> | Alignment Mask | X-AIGD |  | Synthbuster |  | Chameleon |  | CommFor |  |
> |:---:|:---:|:---:|:---:|:---:|:---:|:---:|:---:|:---:|
> |  | Acc | F1 | Acc | F1 | Acc | F1 | Acc | F1 |
> | None ($\beta=0$) | 82.2 | 82.9 | 71.2 | 60.6 | 68.0 | 68.2 | 72.1 | 63.6 |
> | Benign | 80.7 | 81.4 | 70.6 | 60.3 | 65.6 | 66.4 | 71.5 | 63.1 |
> | Random | 81.6 | 82.3 | 69.7 | 57.6 | 66.6 | 67.3 | 72.3 | 64.1 |
> | Saliency | 81.3 | 81.9 | 71.4 | 61.8 | 66.0 | 66.5 | 72.4 | 64.5 |
> | Artifact (ours) | **83.1** | **84.0** | **71.8** | **62.3** | **68.7** | **68.8** | **73.7** | **66.5** |
>
> Notes:
> - The "Random" and "Benign" masks follow the definitions described in our response to Reviewer xCmJ (W2).
> - Due to time constraints, we have not tuned the benign-region weight $\lambda$ or the alignment-loss weight $\beta$, leaving space for further improvements.
>
> The above experiments have been included in **Appendix D.4** of the revision.

---

> ### Author Response · Authors · 2025-11-27
>
> **Q1: Include a more comprehensive comparison with related annotated datasets.**
>
> Thanks. We have expanded Table 1 in the manuscript to provide a more comprehensive comparison across existing annotated datasets:
> | Dataset | Real Images | Fake Images |  | Artifact Annotation |  |  |  |
> |:---:|:---:|:---:|:---:|:---:|:---:|:---:|:---:|
> |  |  | #Image | #Generator | Localization | #Category | #Instance | Annotator |
> | FakeClue | Paired | ~5,000 | Unknown | N/A | N/A | N/A | MLLM |
> | FakeBench | Unpaired | 3,000 | 10 | N/A | 14 | N/A | MLLM + Human |
> | MMFR-Dataset | Paired | 60,266 | Unknown | N/A | 5 | N/A | MLLM |
> | FakeChain | Unpaired | 23,797 | 17 | N/A | 16 | N/A | MLLM + Human |
> | PAL4VST | N/A | 10,168 | 11 | Pixel mask | N/A | 40,841 | Human |
> | SynArtifact | N/A | 1,310 | 8 | Bounding box | 13 | 1,593 | Human |
> | RichHF-18K | N/A | 11,140 | Unknown | Point | N/A | 82,430 | Human |
> | LOKI (image) | Unpaired | 229 | 10 | Bounding box | N/A | 687 | Human |
> | SynthScars | N/A | 12,236 | Unknown | Pixel mask | 3 | 26,566 | Human |
> | X-AIGD (ours) | Paired | 3,337 (labeled) + 48,663 (unlabeled) | 13 | Pixel mask | 7 | 18,202 | Human |
>
> The above table has been included in **Appendix A.4 (Table 8)** of the revision.
>
> ---
>
> **Q2: Discussion on editing-driven automatic construction of annotated data.**
>
> We appreciate this insightful question, which highlights a promising direction for scaling interpretable AIGI detectors through programmatically generated artifact annotations. While synthetic editing-driven masks offer precise control over edited regions, these masks do not necessarily correspond to perceptual artifacts. As a result, detectors trained solely on such data may overfit to editing boundaries or low-level discrepancies between real and edited pixels rather than learning the perceptual artifacts exhibited in fully generated images.
>
> Following the suggestion, we conduct an initial experiment to assess the effectiveness of editing-driven annotations under our proposed attention-alignment loss. We compare three setups of training data:
> - **X-AIGD'**: a subset of X-AIGD using three generators (SD 1.4, SD 3, PixArt-alpha).
> - **Edit-AIGD**: replacing the fake images in X-AIGD' with edited images generated by the corresponding inpainting models; masks are derived from the saliency detector used in Sec. 6.
> - **Hybrid**: a combination of X-AIGD' and Edit-AIGD.
>
> Test accuracy (acc) on four datasets is shown below:
>
> | Training Data | X-AIGD | Synthbuster | Chameleon | CommFor |
> |:---:|:---:|:---:|:---:|:---:|
> | X-AIGD' | 67.2 | 66.4 | 47.7 | 62.3 |
> | Edit-AIGD | 51.4 | 70.1 | 39.2 | 54.3 |
> | Hybrid | **70.6** | **71.0** | **54.5** | **62.9** |
>
> These results indicate that synthetic editing-driven data alone may be insufficient for training robust detectors on fully generated images, likely due to the limited diversity and artificial regularities in edited regions. However, incorporating editing-driven data alongside human-annotated ones yields gains, demonstrating the value of such data as a complementary source when human-labeled samples are limited.
>
> A discussion on scaling up annotated data with automatic or semi-automatic methods has been added to **Appendix E** of the revision.
> We will continue to explore this direction and integrate broader hybrid strategies in future works.
>
> ---
>
> **Q3: Can X-AIGD measurably improve a strong VLM on AJ and PAD?**
>
> Thank you for this valuable question. To investigate, we perform a set of preliminary experiments using GRPO-based reinforcement fine-tuning [a] on Qwen2.5-VL for AJ and PAD, where PAD is treated as an object detection task with IoU-based rewards.
>
> As expected, AJ performance improves: for the 3B model, average accuracy increases from 57.3% to 68.1%. For PAD, fine-tuning yields notable gains only for the Symbols category as measured by the instance-level F1 metric, while improvements for other artifact types remain marginal, even with the 7B model:
> | Model | Textures | Edges & Shapes | Symbols | Color | Semantics | Commonsense | Physics |
> |:---:|:---:|:---:|:---:|:---:|:---:|:---:|:---:|
> | Qwen2.5-VL 7B | 1.3 | 0.3 | 18.8 | 3.0 | 1.3 | 0.0 | 0.0 |
> | Fine-tuned | 1.9 | 1.2 | 31.5 | 1.9 | 3.2 | 2.5 | 2.5 |
>
> These results align with our findings in response to Reviewer S6KQ W1/Q1, indicating that pretrained MLLMs have weak initial competence on most artifact types, limiting the effectiveness of reinforcement fine-tuning.
>
> Overall, these observations indicate that X-AIGD can improve certain artifact categories, although broader gains likely require more effective training strategies and complementary supervision such as the pseudo-labeled synthetic data discussed in Q2. It would be valuable to further study these directions in future work.
>
> The above experiment has been included in **Appendix C.3** of the revision.
>
> [a] Visual-RFT: Visual Reinforcement Fine-Tuning. ICCV 2025.

---

### Official Review · Reviewer_xCmJ · 2025-10-28

**Soundness:** 3
**Presentation:** 2
**Contribution:** 3
**Rating:** 4
**Confidence:** 5

**Summary:**

This paper introduces an interpretable image forgery benchmark that includes pixel-level masks and artifact categories, featuring a fine-grained taxonomy of seven subcategories grouped into three major types, ranging from low-level distortions to high-level semantic or commonsense anomalies. Moreover, the authors demonstrate that existing methods make limited use of perceptible artifacts, relying primarily on high-level, imperceptible cues for discrimination. To address this issue, the authors design an **attention alignment loss** that explicitly guides the model to focus on perceptible artifact regions, achieving superior performance as a result.

**Strengths:**

1. The proposed interpretable benchmark with pixel-level artifact masks is of substantial value to the research community, promoting further progress in explainable image forgery detection and attribution.
2. The paper reveals that current detectors make limited use of perceptible artifacts. By introducing an attention alignment loss, the authors encourage the model to make judgments from a human perceptual perspective, which adds credibility to the interpretability of the approach.

**Weaknesses:**

1. The paper lacks detailed information about whether and when the proposed benchmark will be publicly released. Clear disclosure of dataset availability is essential for reproducibility and community adoption.
2. The claim that existing detectors rely on imperceptible artifacts is indeed intuitive. To further validate the effectiveness of the proposed attention alignment loss, more ablation studies are needed. For example:
   - (1) What happens if the ground-truth mask is randomly assigned?
   - (2) What if attention is forced to align with a normal (artifact-free) object region in the image (which can be segmented using tools such as SAM)?
These experiments would help clarify whether the performance gains truly stem from aligning attention with perceptible artifact regions    rather than from generic spatial regularization effects.

**Questions:**

Please refer to the *Weakness* section for detailed explanations.
If the authors can adequately address my concerns, I would be very willing to raise my score.

---

> ### Author Response · Authors · 2025-11-27
>
> **W1: Lacking detailed information about the public release of the benchmark.**
>
> We will publicly release the generated images, the fine-grained artifact annotations, and all accompanying metadata (such as the confidence scores for each annotation) upon acceptance of the paper. We are committed to ensuring reproducibility and facilitating community adoption of the benchmark.
>
> ---
>
> **W2: More ablation studies are needed for the proposed attention alignment loss.**
>
> Thank you for the insightful suggestions. We conduct the suggested ablations using alternative alignment masks for fake training images:
> - **Random** mask: a randomly sampled convex region within the image;
> - **Benign** mask: the largest SAM-segmented region that contains no overlap with any annotated artifact mask (to avoid noisy small fragments).
>
> We train the models using these masks under the same attention alignment objective and compare them with the proposed artifact-based alignment:
> | Alignment Mask | X-AIGD |  | Synthbuster |  | Chameleon |  | CommFor |  |
> |:---:|:---:|:---:|:---:|:---:|:---:|:---:|:---:|:---:|
> |  | Acc | F1 | Acc | F1 | Acc | F1 | Acc | F1 |
> | None ($\beta=0$) | 82.2 | 82.9 | 71.2 | 60.6 | 68.0 | 68.2 | 72.1 | 63.6 |
> | Benign | 80.7 | 81.4 | 70.6 | 60.3 | 65.6 | 66.4 | 71.5 | 63.1 |
> | Random | 81.6 | 82.3 | 69.7 | 57.6 | 66.6 | 67.3 | 72.3 | 64.1 |
> | Artifact (ours) | **83.1** | **84.0** | **71.8** | **62.3** | **68.7** | **68.8** | **73.7** | **66.5** |
>
> Both Benign and Random masks yield results comparable to or worse than the no-alignment baseline, whereas artifact-based alignment consistently provides the largest improvements across datasets. These findings indicate that the performance gains indeed stem from aligning attention with perceptual artifact regions rather than generic spatial regularization effects.
>
> The above experiments have been included in **Appendix D.4** of the revision.

---

### Official Review · Reviewer_S6KQ · 2025-10-29

**Soundness:** 3
**Presentation:** 2
**Contribution:** 3
**Rating:** 6
**Confidence:** 4

**Summary:**

The paper introduces X-AIGD, a high-quality, fine-grained benchmark for interpretable AI-generated image detection. It provides pixel-level, multi-level  artifact annotations on paired real-fake images generated from 13 state-of-the-art text-to-image models. Using this benchmark, the authors systematically reveal that existing AIGI detectors barely rely on human-perceptible artifacts, even at the lowest distortion level. They further explore artifact detection as an auxiliary task and attention alignment, showing that explicitly grounding model attention to annotated artifact regions significantly improves both interpretability and cross-model generalization.

**Strengths:**

- X-AIGD introduces the first benchmark with paired real-fake images and pixel-level annotations across 7 fine-grained artifact categories, spanning low-level distortions, high-level semantics, and cognitive-level counterfactuals, enabling grounded interpretability evaluation.

- Empirical analysis surprisingly reveals that SOTA detectors achieve good performance without relying on perceptual artifacts—contrary to common intuition that visible flaws should be primary cues, as evidenced by uncorrelated performance across fidelity metrics in Fig. 3 and severely misaligned heatmaps in Fig. 4.

- Attention alignment to ground-truth artifact regions significantly improves both interpretability and cross-generator generalization, providing a practical and effective method to make AIGI detection more human-trustworthy.

**Weaknesses:**

- No comparison or integration with MLLM-based explanation methods (e.g., GPT-4o, LLaVA) is provided, despite critiquing their lack of grounding—missing an opportunity to show whether X-AIGD enables better spatial reasoning in language models.

- Inter-annotator agreement (e.g., IoU-based Fleiss’ κ) is not reported, especially critical for subjective high-level and cognitive-level artifacts, potentially affecting reproducibility.

**Questions:**

Could the authors provide the results of MLLM-based explanation methods?

---

> ### Author Response · Authors · 2025-11-25
>
> **W1/Q1: Evaluation and integration of MLLM-based explanation methods.**
>
> Thank you for the valuable comment. Although our work focuses on expert AIGI detectors, examining MLLM-based explanation methods is valuable. Below, we provide some analyses based on X-AIGD.
>
> **(1) Evaluating MLLMs’ ability to detect perceptual artifacts.**
> We separately prompt an MLLM to identify each artifact category in a test fake image and return bounding boxes for detected instances, if any. We observe that most predicted boxes are highly imprecise, despite the models’ general grounding capability on other high-level tasks. To obtain a meaningful measure, we evaluate only image-level binary predictions: any non-empty bounding-box list counts as predicting the presence of that artifact type. The image-level F1-scores across the seven artifact categories are shown below:
> | Model | Textures | Edges & Shapes | Symbols | Color | Semantics | Commonsense | Physics |
> | :---- | :------: | :----: | :-----: | :---: | :-------: | :---------: | :-----: |
> | Qwen2.5-VL 7B  |   5.0    |      5.3       |  49.0   | 14.6  |   26.7    |    11.7     |  14.3   |
> | InternVL3 8B  |   16.2   |      38.6      |  47.2   | 13.2  |   39.6    |    18.7     |  13.4   |
> | LLaVa-Next (Mistral-7B) |   32.9   |      69.1      |  28.7   |  9.0  |   52.6    |    17.1     |  12.2   |
> | DeepSeek-VL2-Small  |   32.3   |      71.1      |  33.2   |  7.8  |   45.2    |    18.5     |  17.1   |
> | GPT-4o  |   9.3    |      59.5      |  91.7   |  0.0  |   40.5    |    32.3     |   9.1   |
> | **Random**  |   29.9   |      56.7     |  30.6   |  9.4  |   49.1    |    19.2     |   12    |
>
> Across most categories, the MLLMs’ predictions do not exceed random guessing, with the notable exception of the Symbols category. This suggests that current MLLMs may lack the fine-grained perceptual sensitivity required for grounded artifact reasoning.
>
> **(2) Improving the explanation ability of MLLMs**
> We also conduct a preliminary experiment by applying GRPO-based reinforcement fine-tuning (following [a]) to Qwen2.5-VL-7B on our PAD task, treating artifact detection as object detection and using IoU-based rewards. The table below shows the instance-level F1-scores at threshold $t=0.5$ (please refer to Appendix A.5 of the revision for its definition) for the model before and after fine-tuning:
> | Model | Textures | Edges & Shapes | Symbols | Color | Semantics | Commonsense | Physics |
> |:---:|:---:|:---:|:---:|:---:|:---:|:---:|:---:|
> | Qwen2.5-VL 7B | 1.3 | 0.3 | 18.8 | 3.0 | 1.3 | 0.0 | 0.0 |
> | Fine-tuned | 1.9 | 1.2 | 31.5 | 1.9 | 3.2 | 2.5 | 2.5 |
>
> The GRPO fine-tuning leads to a clear improvement only for the Symbols category, while other artifact types showed marginal gains. This aligns with our finding in (1) that pretrained MLLMs exhibit weak initial competence on most artifact types, making reinforcement fine-tuning ineffective. We expect future works to explore more effective training paradigms and scale up the training data.
>
> The above experiments and discussions have been included in **Apendix C** in the revision.
>
> [a] Visual-RFT: Visual Reinforcement Fine-Tuning. ICCV 2025.
>
> ---
>
> **W2: Inter-annotator agreement is not reported.**
>
> We clarify that our annotation protocol uses a relay-style incremental workflow rather than three fully independent passes, so inter-annotator agreement is not reported. Each subsequent annotator is instructed to review the existing annotations, identify any missing artifacts, and supplement new labels when necessary. This workflow prioritizes annotation completeness over strict consensus.
>
> We acknowledge that artifact annotation is inherently subjective and could lead to unavoidable disagreements. Instead of enforcing consensus, we retain all annotations from the three annotators and subsequently conduct an independent confidence-scoring process, which yields the aggregated statistics shown in Fig. 2. In the final revision, we (1) extend confidence scoring to all annotations and (2) release per-annotator confidence scores as part of the metadata. This allows future work to filter or weight annotations according to their confidence levels.
>
> The clarification on our annotation workflow has been updated in **Appendix A.3** of the revision, and additional discussions on mitigating the impact of inter-annotator variance are included in **Appendix A.6**.

---

### Official Review · Reviewer_PcLD · 2025-10-31

**Soundness:** 3
**Presentation:** 3
**Contribution:** 3
**Rating:** 6
**Confidence:** 5

**Summary:**

The paper presents X-AIGD, a benchmark for interpretable AI-generated image detection. The dataset pairs real and generated images and provides pixel-level masks of perceptual artifacts. Using this benchmark, the authors show that current detectors, despite strong real/fake accuracy, often do not actually focus on these artifact regions. To improve this, they add an artifact segmentation auxiliary task and an attention alignment loss. These changes yield more interpretable predictions and better cross-dataset generalization.

**Strengths:**

- The dataset contribution is concrete and timely, providing pixel-level artifact masks with semantic labels and paired real/fake images. This directly supports studying why a detector made its decision instead of treating the task as only real or fake.
- The analysis of existing detectors is systematic. The authors compare saliency and attention maps with ground-truth artifact masks and show that they often do not overlap, which raises doubts about how explainable current high-performing detectors actually are.
- The paper appropriately introduces perceptual artifact segmentation as an auxiliary task and constrains the model to focus on artifact regions through attention alignment. This approach leads to more stable F1 scores across datasets and more consistent visual explanations, directly addressing the limitations of prior detectors.
- The writing quality is strong, and the logic of the paper is very clear and easy to follow.

**Weaknesses:**

- There is a growing line of work that uses multimodal large language models (MLLMs) to "look at an image and generate textual explanations," and claims to identify "where it is fake and why it is fake." The authors briefly mention related efforts (e.g., using MLLMs to generate explanations), but a more explicit discussion of works such as [1,2,3] would make the connection to this direction clearer for readers and better contextualize the contribution.
- The authors report 12 human annotators, 3 annotation rounds, and subjective confidence scores, indicating high pixel-level quality. However, the full-resolution / region-level protocol is extremely expensive, and ultimately only ~3k forged samples received high-quality annotations. This cost is understandable, but it raises a concern: is the pipeline realistically reproducible or extensible for larger-scale data? The paper would be stronger with a discussion of scalability and reproducibility, and possible ways to reduce cost (e.g., using MLLMs to assist or pre-label forged regions).
- **Typos**: On line 52-53, “X-AIGD (eXplainable AI-**Genereted** image Detection)” should be changed to “X-AIGD (eXplainable AI-**Generated** Image Detection)”

[1] Heie: Mllm-based hierarchical explainable aigc image implausibility evaluator

[2] Spot the fake: Large multimodal model-based synthetic image detection with artifact explanation

**Questions:**

See Weaknesses

---

> ### Author Response · Authors · 2025-11-25
>
> **W1: A more explicit discussion of MLLM-based interpretable AIGI detection methods should be presented.**
>
> Thank you for the valuable suggestion. During the rebuttal, we conducted a focused evaluation of three open-source representatives of this line of work, namely FakeVLM [a], FakeReasoning [b], and LEGION [c], using the X-AIGD benchmark.
>
> **(1) Authenticity judgment performance.**
>
> The textual output of FakeVLM and FakeReasoning comprises a binary authenticity prediction and accompanying explanations. When evaluated on X-AIGD, both models exhibit significant **class bias**, resulting in low average accuracy:
> | Model | Real Acc. | Fake Acc. | Avg. Acc. |
> | ---- | ---- | ---- | ---- |
> | FakeVLM | 43.72 | 98.93 | 71.33 |
> | FakeReasoning | 81.16 | 39.69 | 60.43 |
>
> In addition, LEGION produces a non-zero artifact mask and at least one artifact description for **every** input image, including real ones. This behavior can be explained by the fact that LEGION is trained exclusively on fake images (SynthScars) and never observes real image samples.
>
> **(2) Qualitative assessment of explanations.**
>
> Our qualitative analysis highlights several recurring issues:
> - FakeVLM and FakeReasoning: Often rely on vague or ungrounded judgments such as "inconsistent light source", "lacks realistic detail", or "(objects) do not match reality". While some common cues (e.g., "overly smooth", "saturation is too high") are correct in some cases, they frequently appear as false positives. These patterns likely stem from the MLLMs used for generating the annotations for their training data.
> - LEGION: While its binary masks can localize some artifact regions, the generated textual explanations may be inconsistent with the predicted masks. We observe hallucinated object parts (e.g., nonexistent hands) or descriptions of abnormalities that do not correspond to any detected region, suggesting potential overfitting to textual patterns rather than grounded visual evidence.
>
> **(3) Quantitative evaluation of artifact localization.**
>
> We evaluate LEGION’s artifact masks on the Category-Agnostic PAD task. Compared with the results taken from Table 13 in our Appendix, LEGION outperforms a vanilla fine-tuned DINOv2 baseline in all metrics but remains inferior to our attention-alignment model trained on our annotated data:
> | **Model**  | **IoU**  | **PixP**  | **PixR**  | **PixF1** |
> | ------ | ---- | ----- | ----- | ----- |
> | Baseline | 0.5 | 24.4 | 0.5 | 1.0 |
> | LEGION | 9.1 | 29.1 | 11.7 | 16.7 |
> | Ours (Attention Alignment) |14.0 | 37.2 | 18.2 | 24.5 |
>
> **Summary.**
>
> These findings highlight the importance of X-AIGD. Existing MLLM-based methods may generate explanations that are ungrounded, inconsistent, or biased, and they lack access to paired real and generated images with fine-grained human annotations that are essential for reliable interpretability evaluation. X-AIGD addresses these limitations by offering localized and trustworthy supervision, which enables principled analysis of whether models truly attend to perceptual artifacts rather than spurious cues.  Our revision has cited and discussed these representative works in **Appendix C**.
>
> [a] Spot the Fake: Large Multimodal Model-Based Synthetic Image Detection with Artifact Explanation. NeurIPS 2025.
> [b] Towards Generalizable Forgery Detection and Reasoning. Arxiv 2025.
> [c] LEGION: Learning to Ground and Explain for Synthetic Image Detection. ICCV 2025.

---

> ### Author Response · Authors · 2025-11-25
>
> **W2: Scalability of the construction of labelled data.**
>
> Thank you for raising this point. While pixel-level and category-specific perceptual artifact annotation is costly, it is feasible to scale the pipeline by adopting more efficient strategies:
>
> A common practical path is **semi-automatic expansion**: detectors trained on X-AIGD can generate coarse artifact proposals, and annotators only refine uncertain regions. This could reduce manual effort while preserving annotation quality if an effective detector can be constructed and continuously refined.
>
> Another option is **pseudo-labeling via controlled image editing**, as suggested by Reviewer NFjz. By editing real images with predefined masks, the masks can serve as pseudo artifact regions. However, edited regions may not always correspond to perceptual artifacts, and models risk exploiting editing boundaries or low-level distinctions between real and fake (edited) pixels rather than genuine cues. Careful design of image processing and training strategies is needed.
> As a preliminary investigation, we have generated a batch of such data and made an attempt to train models with it. Please refer to our response to Q2 of Reviewer NFjz.
>
> We have included the above discussion in **Appendix E** of the revision.
>
> ---
>
> **W3: Typo on line 52-53.**
>
> Thank you for your meticulous comment. We have double-checked the manuscript and corrected the typos accordingly.

---

### Meta-Review · Area_Chair_Q5NB · 2026-01-05

**Summary:**

This submission introduces X-AIGD, a fine-grained benchmark for interpretable AI-generated image (AIGI) detection with pixel-level artifact annotations, addressing a critical gap in existing AIGI detection research, i.e., the lack of grounded interpretability.

Reviewers uniformly recognized the work’s timely and concrete contributions, including the well-structured X-AIGD benchmark, systematic analysis of detector reliance on perceptual artifacts, and effective attention-alignment approach. They also raised constructive concerns spanning MLLM-based method evaluation, inter-annotator variation, dataset comparison, ablation studies, and annotation scalability. The authors provide a satisfactory response to address the problems.

**Reviewer Scores:**

Reviewer xCmJ would have changed the score if they had been able to participate fully in the discussion.

---

### Decision · Program_Chairs · 2026-01-26

Accept (Poster)